# Optogenetic control shows that kinetic proofreading regulates the activity of the T cell receptor

O Sascha Yousefi[1,2,3], Matthias Günther[4,5], Maximilian Hörner[1,2], Julia Chalupsky[1,2,6], Maximilian Wess[1,2], Simon M Brandl[1,2], Robert W Smith[7], Christian Fleck[7], Tim Kunkel[2], Matias D Zurbriggen[1,2,8], Thomas Höfer[4,5], Wilfried Weber[1,2], Wolfgang WA Schamel[1,2,7]*

[1]Signalling Research Centres BIOSS and CIBSS, University of Freiburg, Freiburg, Germany; [2]Faculty of Biology, University of Freiburg, Freiburg, Germany; [3]Spemann Graduate School of Biology and Medicine, University of Freiburg, Freiburg, Germany; [4]Division of Theoretical Systems Biology, German Cancer Research Center, Heidelberg, Germany; [5]BioQuant Center, University of Heidelberg, Heidelberg, Germany; [6]Center for Chronic Immunodeficiency, Medical Center Freiburg and Faculty of Medicine, University of Freiburg, Freiburg, Germany; [7]Laboratory of Systems and Synthetic Biology, Wageningen University and Research, Wageningen, Netherlands; [8]Institute of Synthetic Biology and Cluster of Excellence on Plant Sciences, University of Düsseldorf, Düsseldorf, Germany

*For correspondence:
wolfgang.schamel@biologie.uni-freiburg.de

Competing interests: The authors declare that no competing interests exist.

**Abstract** The immune system distinguishes between self and foreign antigens. The kinetic proofreading (KPR) model proposes that T cells discriminate self from foreign ligands by the different ligand binding half-lives to the T cell receptor (TCR). It is challenging to test KPR as the available experimental systems fall short of only altering the binding half-lives and keeping other parameters of the interaction unchanged. We engineered an optogenetic system using the plant photoreceptor phytochrome B (PhyB) as a ligand to selectively control the dynamics of ligand binding to the TCR by light. This opto-ligand-TCR system was combined with the unique property of PhyB to continuously cycle between the binding and non-binding states under red light, with the light intensity determining the cycling rate and thus the binding duration. Mathematical modeling of our experimental datasets showed that indeed the ligand-TCR interaction half-life is the decisive factor for activating downstream TCR signaling, substantiating KPR.
DOI: https://doi.org/10.7554/eLife.42475.001

## Introduction

The function of T cells is to mount an immune response to foreign ligands, such as derived from bacteria or viruses, but not to respond to self ligands stemming from the body's own cells. These ligands are composed of a foreign peptide presented by major histocompatibility complexes molecules (pMHC) on the own cells. Activation of a T cell is initiated when foreign pMHC bind to the T cell receptor (TCR) on the T cell surface. The pMHC-TCR binding event stimulates intracellular signaling pathways, such as calcium influx into the cytosol, leading to the functional responses of the T cell (*Courtney et al., 2018*). Self peptides on MHC (self pMHCs) also bind to the TCR and are important for the development and survival of naïve T cells, but do not trigger an immune response as seen for foreign peptides on MHC (*Davis et al., 1998*). This discrimination between foreign and self pMHC correlates with the affinity of the ligand-TCR interaction, in that foreign, stimulatory pMHCs bind

with higher affinity to the TCR than non-stimulatory pMHC (*Davis et al., 1998*; *Sykulev et al., 1994*). However, how the affinity of a ligand is determined by the cell to generate a T cell response or not remains enigmatic (*Chakraborty and Weiss, 2014*). Note that in case of pMHC binding to T cells other processes than the pure pMHC-TCR interaction are involved, such as interactions with the co-receptors CD8 or CD4; thus, the terms 'apparent affinity' or 'potency' might be more suitable when describing these complex binding events.

One model is kinetic proofreading (KPR), which originally described the specificity by which the genetic code is read in protein synthesis (*Hopfield, 1974*) and inspired a similar theoretical model for ligand discrimination in T cells (*McKeithan, 1995*). In KPR the T cell does not simply measure the amount of ligand-bound TCRs (called occupancy model), but monitors the dynamics of the binding events. These dynamics can be described by the on-rate and the half-life of the interaction. The KPR model proposes that a long half-life of the ligand-TCR interaction, such as seen for high affinity pMHC, allows a series of biochemical reactions to be completed that eventually trigger downstream signaling. By contrast, a low affinity ligand detaches before an activatory signal is produced and the TCR then reverts quickly to the initial inactive state, thus not initiating T cell activation. Although the half-life is the decisive factor, it was recently shown that the on-rate also plays a role (*Aleksic et al., 2010*; *Govern et al., 2010*; *Lin et al., 2019*). If the on-rate is very fast a ligand that has detached can rapidly rebind to the same TCR before the first biochemical reactions are reverted. Again, the duration of the binding event, in this case interrupted by short dissociations, is the relevant parameter.

The KPR model has also been extended to include feedback and feed-forward loops in the signaling network below the TCR (*Altan-Bonnet and Germain, 2005*; *Chakraborty and Weiss, 2014*; *Dushek et al., 2011*; *Lever et al., 2016*; *Rabinowitz et al., 1996*). Inclusion of these signaling network loops improved the mathematical description of the observed sharp ligand discrimination threshold, when relating ligand half-life to T cell activation. At the same time, the high sensitivity of the T cells towards low numbers of ligands (1–10 molecules) was retained (*Irvine et al., 2002*; *Purbhoo et al., 2004*).

To get experimental insight into the mechanism of ligand discrimination by T cells, pMHC or TCRs have been mutated at the binding sites to generate ligand-TCR pairs of different affinities and half-lives (*Aleksic et al., 2010*; *Altan-Bonnet and Germain, 2005*; *Daniels et al., 2006*; *Davis and van der Merwe, 2006*; *Dushek et al., 2011*; *Govern et al., 2010*; *Holler and Kranz, 2003*; *Kalergis et al., 2001*; *Kersh et al., 1998*; *Krogsgaard et al., 2003*; *Lever et al., 2016*). Although such studies are broadly consistent with KPR, other biophysical parameters, such as the free binding energy, geometry of the interaction (*Adams et al., 2011*), conformational changes at the TCR (*Dopfer et al., 2014*; *Gil et al., 2002*; *Risueño et al., 2006*) and the ability to withstand pulling (*Kim et al., 2009*; *Liu et al., 2014*), might also have been changed along with the affinity, and therefore alternative models of ligand discrimination cannot be ruled out. Unfortunately, no method to specifically modulate only the dynamics of ligand-receptor interactions is currently available. Thus, in order to disentangle the half-life from these other parameters, we engineered an optogenetic system in which the duration of ligand binding to the TCR can be remotely controlled in a reversible manner (ON-OFF switch), called the opto-ligand-TCR system.

Our opto-ligand-TCR approach harnesses the PhyB-PIF (phytochrome B-PhyB interacting factor) protein pair from *Arabidopsis thaliana* (*Bae and Choi, 2008*; *Levskaya et al., 2009*; *Toettcher et al., 2013*). In this pair, the photoreceptor PhyB is the light-responsive element, due to its chromophore phycocyanobilin, which undergoes a conformational cis-trans isomerization when absorbing photons of the appropriate wavelength. Upon illumination with 660 nm light, PhyB switches to its ON state in which it interacts with PIF6 with a nanomolar affinity (*Levskaya et al., 2009*). With 740 nm light, PhyB undergoes a conformational transition to the OFF state preventing binding to PIF6. This light-dependent protein-protein interaction was utilized in several optogenetic applications (*Kolar et al., 2018*), such as the control of protein or organelle localization (*Adrian et al., 2017*; *Beyer et al., 2018*; *Levskaya et al., 2009*), intracellular signaling (*Toettcher et al., 2013*), nuclear transport of proteins (*Beyer et al., 2015*), cell adhesion (*Baaske et al., 2019*; *Yüz et al., 2018*) or gene expression (*Müller et al., 2013a*). Using high intensity light, the PhyB-PIF interaction can be switched ON and OFF within seconds (*Levskaya et al., 2009*; *Mancinelli, 1994*; *Smith et al., 2016*). Importantly for our study, at continuous 660 nm illumination the individual PhyB molecules constantly switch between the ON and OFF states, again in the

order of seconds, thus being within the range of the estimated KPR times (*Mancinelli, 1994*; *Smith et al., 2016*).

We and others have previously fused binding domains to the ectodomain of the TCRβ subunit; either a single chain Fv fragment (*Minguet et al., 2007*) or a single strand DNA oligonucleotide (*Taylor et al., 2017*). Indeed, the chimeric TCRs were expressed on the cell surface and were activated via the appended binding domains. Importantly, ligand discrimination also occurred when using the DNA-TCR; i.e., a low affinity binder to the DNA did not evoke TCR stimulation and a high affinity binder did (*Taylor et al., 2017*). This clearly showed that ligands do not need to bind to the canonical pMHC binding site within the TCR and that co-receptors are not required for ligand discrimination. It should be noted that the developmental state of the T cell can modulate the discrimination process as do the co-receptors (CD8 or CD4) or the expression levels of intracellular signaling molecules (*Altan-Bonnet and Germain, 2005*; *Davey et al., 1998*; *Lucas et al., 1999*; *Madrenas et al., 1997*; *Stepanek et al., 2014*).

Here we fused the first 100 amino acids of PIF6 together with GFP to the ectodomain of TCRβ and used the first 651 amino acids of PhyB in a tetramerized form as the ligand (*Figure 1*). Using continuous 660 nm light of different intensities to modulate the dynamics of PhyB tetramer binding to the TCR and calcium influx as a readout we find that there is an intensity threshold: at lower intensities and longer ligand-TCR half-lives the T cell is activated and at higher intensities and shorter half-lives the cell is not activated. Using a mathematical model of KPR we show that the threshold half-life in our opto-ligand-TCR system is 8 s.

## Results

The first aim of our study was to establish an optogenetic system in which ligand binding to the TCR can be reversibly controlled by light (*Figure 1*).

### Engineering of the opto-ligand-TCR system: the ligand

The light-responsive N-terminal 651 amino acids of *A. thaliana* PhyB (PhyB$_{1-651}$) have been used as an optogenetic tool (*Adrian et al., 2017*; *Baaske et al., 2019*; *Beyer et al., 2015*; *Beyer et al., 2018*; *Johnson and Toettcher, 2018*; *Levskaya et al., 2009*; *Müller et al., 2013b*; *Toettcher et al., 2013*) and the photobiology of this fragment has been described previously (*Smith et al., 2016*). Here we used this PhyB form as a ligand. PhyB$_{1-651}$ fused to the biotinylation site Avitag (*Beckett et al., 1999*) and a His$_6$-tag (*Figure 2A*) was produced in *E. coli*. Additionally, the bacteria were engineered to produce the cyanobacterial version of the phytochrome chromophore, phycocyanobilin (*Essen et al., 2008*; *Smith et al., 2016*). PhyB$_{1-651}$-Avitag-His$_6$, called PhyB in the remainder of this article, was isolated by Ni$^{2+}$-affinity chromatography (*Smith et al., 2016*). We then tested the functionality of PhyB through its light-dependent interaction with PIF6. To this aim, we produced the first 100 amino acids of *A. thaliana* PIF6 (PIF6$_{1-100}$), which were shown to be sufficient for photoreversible PhyB binding with nanomolar affinity (*Tischer and Weiner, 2014*), as a fusion protein with the maltose-binding protein and a His$_6$-tag [MBP-PIF6$_{1-100}$-His$_6$, from now on called MBP-PIF(wt)]. After illuminating a mixture of PhyB and an excess MBP-PIF(wt) with saturating 660 nm light, 70% of the PhyB molecules were complexed with PIF as depicted by a shift in elution from a size exclusion chromatography column

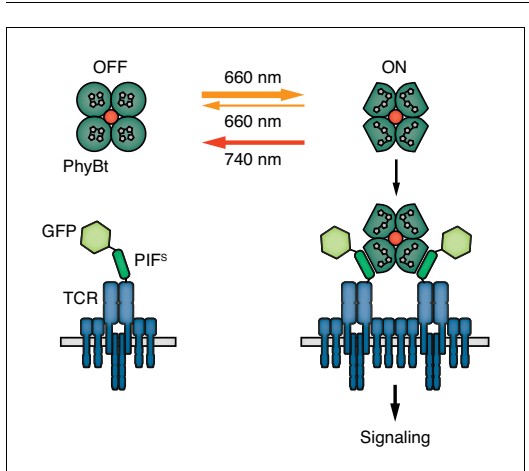

**Figure 1.** Engineering a light-controlled switch for the ligand-TCR interaction: the opto-ligand-TCR system. Light of 660 nm and 740 nm wavelength reversibly switches PhyB between the OFF and ON states. In the ON state PhyB tetramers (PhyBt) bind to and cluster GFP-PIF$^S$-TCRs leading to signaling and the activation of the T cell. The red dot indicates the fluorophore-coupled streptavidin tetramer.
DOI: https://doi.org/10.7554/eLife.42475.002

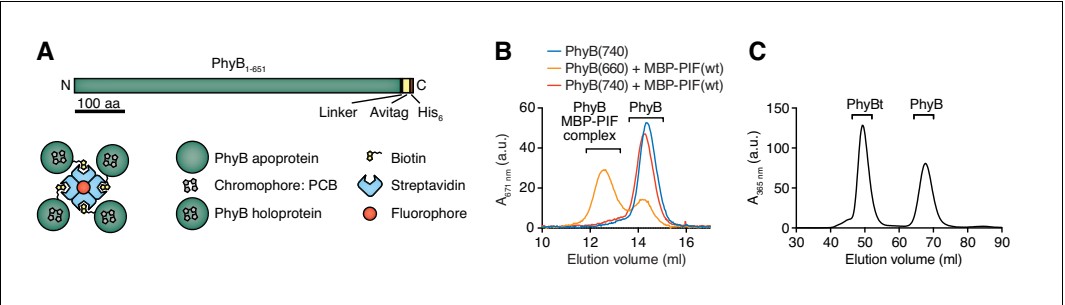

**Figure 2.** Production of PhyB tetramers. (A) Schematics of the PhyB$_{1-651}$ construct and the PhyB tetramers. PCB = phycocyanobilin. (B) Purified PhyB was illuminated with 660 nm light [PhyB(660)] and added in a 1:2 molar ratio to MBP-PIF(wt). The proteins were separated by gel filtration and PhyB was detected by its absorbance at the isosbestic point of 671 nm. PhyB molecules around 14.5 ml elution volume are the free PhyB molecules and the ones around 12.5 ml are the PhyB-MBP-PIF complexes. As controls, PhyB illuminated with 740 nm light [PhyB(740)] plus MBP-PIF(wt) and PhyB alone was only detected at 14.5 ml elution volume. Results show one experiment of n > 3. (C) Affinity chromatography-purified PhyB was mixed in a 10:1 molar ratio with streptavidin-DyLight650, incubated for 2 hr at room temperature and the formed PhyB tetramers (PhyBt) were isolated from monomers using size-exclusion chromatography. The elution of PhyB was monitored via its absorbance at 365 nm. Results show one experiment of n > 3.

DOI: https://doi.org/10.7554/eLife.42475.003

(*Figure 2B*). This was not the case when the proteins were exposed to 740 nm light. Since at photoequilibrium under 660 nm light only 80% of the PhyB molecules are in the ON state (*Bae and Choi, 2008*; *Smith et al., 2016*), we conclude that the majority of PhyB molecules were functionally active.

Although soluble TCR ligands are active as dimers (*Boniface et al., 1998*; *Cochran et al., 2000*; *Minguet and Schamel, 2008*; *Minguet et al., 2007*), tetrameric pMHC based on streptavidin are routinely used to stimulate the TCR (*Altman et al., 1996*) and to obtain insight into ligand discrimination by T cells (*Stone et al., 2011*; *Stone et al., 2001*). Thus, we wanted to construct PhyB tetramers (PhyBt) to be used as ligands in our system (*Figure 1*). To this end, biotinylated PhyB was tetramerized using fluorophore-coupled streptavidin. After separating the tetramers from monomers by size exclusion chromatography (*Figure 2C*), we obtained purified PhyBt that we used in this work.

## Engineering of the opto-ligand-TCR system: the TCR

Next, we engineered a PIF-fused TCR that can bind to and be activated by PhyBt when the PhyB molecules are in the ON (but not in the OFF) state (*Figure 1*). In plants PIF6 is produced in the cytoplasm, whereas in our system PIF6 is produced in the oxidative environment of the endoplasmic reticulum. Therefore, we mutated cysteines and N-linked glycosylation sites (Asn-X-Ser/Thr) in PIF6. We produced a panel of five different PIF6$_{1-100}$ mutants abolishing cysteines 9 and 10 as well as asparagine 35 or serine 37 as MBP fusion proteins (*Figure 3—figure supplement 1A,B*). We analyzed the interaction of PhyB with these PIF6$_{1-100}$ mutants under limiting amounts of MBP-PIF using size exclusion chromatography (*Figure 3—figure supplement 1C,D*). All mutants formed complexes with PhyB pre-illuminated with 660 nm light [PhyB(660)] similar to MBP-PIF(wt).

Having seen that all PIF6$_{1-100}$ mutants interacted well with PhyB, they were fused - preceded by a signal peptide - to the N-terminus of the human HA1.7 TCRβ chain that contains a Vβ3 variable region (*Hennecke et al., 2000*; *Hewitt et al., 1992*) (*Figure 3A*). We analyzed the presence of the different PIF6$_{1-100}$-TCRβ constructs on the cell surface following lentiviral transduction of Jurkat T cells (*Abraham and Weiss, 2004*). PIF6$_{1-100}$ C9S C10S S37A [PIF(SSA)] showed the highest surface presence (*Figure 3B*), indicating that it assembled to a complete TCR complex (*Alarcón et al., 2003*; *Call and Wucherpfennig, 2005*). Hence, PIF(SSA) was therefore used for all future optimizations and termed secretory PIF or PIF$^S$ (*Figure 3C*). Surprisingly, despite the good interaction of MBP-PIF$^S$ with PhyB in size-exclusion chromatography (*Figure 3—figure supplement 1C,D*), no binding of PhyBt to the PIF$^S$-TCR on the surface of Jurkat cells could be detected (*Figure 3D*). GFP-PIF$^S$-TCR cells (described below) served as a positive control for binding (*Figures 3D* and *4F* panels are from the same experiment). Furthermore, PIF$^S$-TCR Jurkat cells could be stimulated to flux

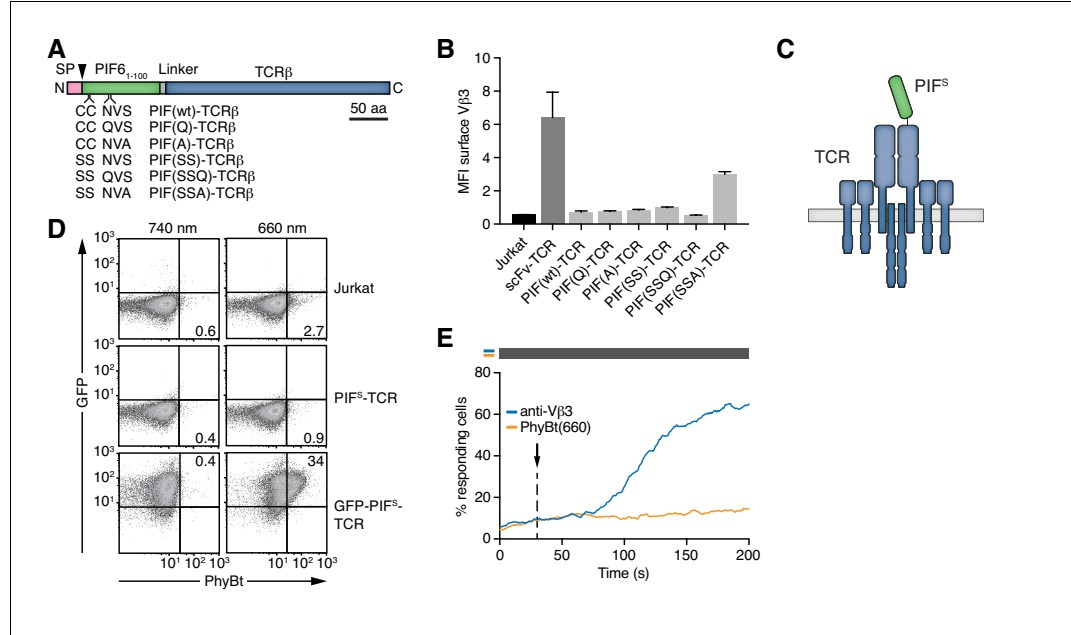

**Figure 3.** Selection of PIF[S] that can be expressed together with the TCR. (**A**) Schematics of the PIF-TCRβ constructs, including wild-type and mutant PIF. SP depicts the signal peptide and the arrow the signal peptidase cleavage site. The schematic constructs are drawn to scale with the scale bar indicated. (**B**) The presence of the different PIF-TCRs and a single chain variable fragment (scFv)-TCR on the cell surface was measured in lentivirally transduced Jurkat cells together with the parental cell line using an anti-Vβ3 antibody (Jovi3) via flow cytometry. The median fluorescence intensity (MFI) averaged for three experiments ± SEM is depicted. (**C**) Scheme of PIF[S]-TCRβ as integrated into the TCR. (**D**) 100 nM phycoerythrin (PE)-labeled PhyBt pre-illuminated with 660 nm or 740 nm light were incubated with Jurkat, PIF[S]-TCR Jurkat and GFP-PIF[S]-TCR cells and binding detected by flow cytometry. Numbers depict the % of cells in the respective quadrant. Results show one experiment of n = 3. (**E**) PIF[S]-TCR cells were labeled with Indo-1 and calcium influx measured by flow cytometry. 100 nM PhyBt(660) (orange) or 1 μg/ml anti-Vβ3 antibody (blue) were added as stimuli. Their addition is marked by an arrow and the illumination procedure by a bar above the graph (grey = dark). Results show one experiment of n > 3.
DOI: https://doi.org/10.7554/eLife.42475.004

The following figure supplement is available for figure 3:

**Figure supplement 1.** Mutations of the Cys and N-linked glycosylation site in PIF6.
DOI: https://doi.org/10.7554/eLife.42475.005

---

calcium via cross-linking of the PIF[S]-TCR using an anti-Vβ3 antibody, but not using PhyBt pre-illuminated with 660 nm light, called PhyBt(660) (**Figure 3E**). Consequently, although PIF[S]-TCRβ is present at the cell surface and PIF[S] itself binds to PhyB (in the form of MBP-PIF), PIF[S] loses its binding capacity towards PhyB when it is fused to the TCR and exposed on the T cell's surface.

A major difference between the functional MBP-PIF[S] and the dysfunctional PIF[S]-TCRβ construct is the C- and N-terminal localization of PIF[S], respectively. Thus, adding an unrelated protein to the N-terminus of PIF[S] might rescue the PhyB-binding ability of the PIF[S]-TCR. To test this possibility, we attached a monomeric green fluorescent protein optimized for the oxidative environment of the endoplasmic reticulum (moxGFP, (*Costantini et al., 2015*)) to the N-terminus of PIF[S]-TCRβ. We distinguished the effect of a permanently attached moxGFP or a moxGFP that is only present during folding of PIF[S] in the endoplasmic reticulum. To this end, we added different furin protease recognition sequences (F1-F3) or a flexible linker without protease cleavage site (noF) between moxGFP and PIF[S] (**Figure 4A**). The protease furin is expressed in the Golgi and would cleave off the moxGFP as the engineered TCRs are exported to the cell surface. All constructs were well expressed on Jurkat cells (**Figure 4B**) and showed the expected absence or presence of moxGFP on the cell surface (**Figure 4C**). The construct using a truncated furin site (F3) had intermediate surface moxGFP levels, indicating that moxGFP is inefficiently cleaved. PhyBt(660) hardly bound to the surface of Jurkat cells expressing GFP-F1-PIF[S]-TCR or GFP-F2-PIF[S]-TCR with efficiently cleaved moxGFP

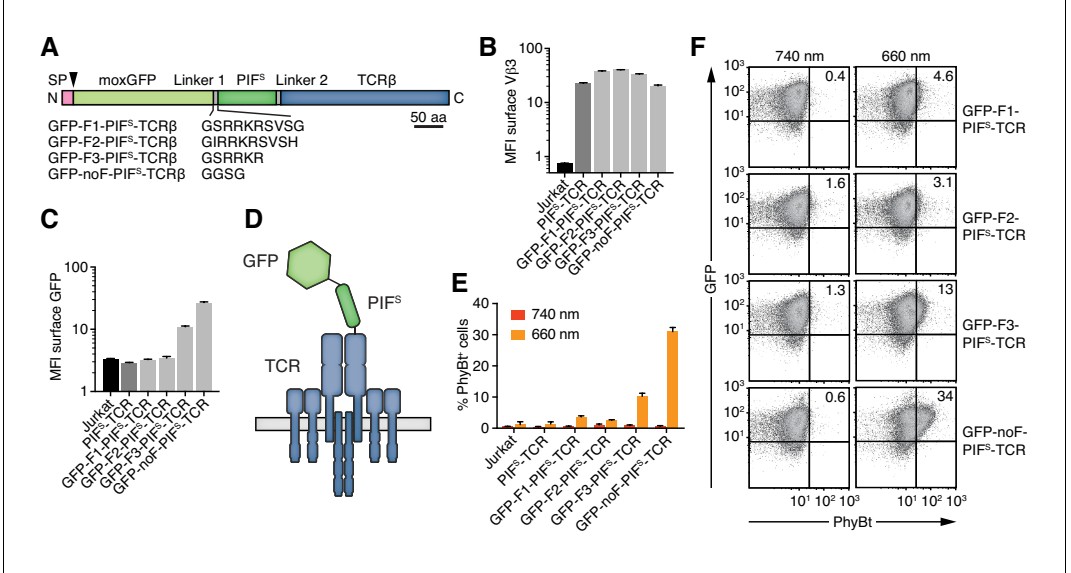

**Figure 4.** Engineering of the GFP-PIF[S]-TCR. (**A**) Schematics of the GFP-PIF[S]-TCRβ constructs, including three different furin cleavage sites (F1, F2, F3) or omitting any cleavage site (noF). SP depicts the signal peptide, the arrow the signal peptidase cleavage site and moxGFP the monomeric GFP optimized for an oxidative environment. (**B**) The surface expression of the different GFP-PIF[S]-TCRs and PIF[S]-TCR was measured in transduced Jurkat cells together with the parental cell line using an anti-Vβ3 antibody (Jovi3) via flow cytometry. (**C**) Analogous to (**B**), the amount of GFP was quantified on the surface of the different transductants using a polyclonal anti-GFP antibody via flow cytometry. (**B**) and (**C**) depict the median fluorescence intensity (MFI) averaged for three measurements ± SEM. (**D**) Scheme of GFP-PIF[S]-TCRβ as integrated into the TCR. (**E**) 100 nM phycoerythrin (PE)-labeled PhyBt pre-illuminated with 660 nm or 740 nm light were incubated with the cells indicated and binding was detected by flow cytometry. One experiment out of three is depicted displaying the average of quadruplicates ± SEM. (**F**) Together with *Figure 3D* these are the GFP vs PhyBt plots of the experiment quantified in (**E**).

DOI: https://doi.org/10.7554/eLife.42475.006

(*Figure 4E,F*). However, fusing moxGFP permanently to PIF[S]-TCRβ resulted in strong light-dependent binding of PhyBt to the cell surface. In line with this, GFP-F3-PIF[S]-TCR with partly cleaved GFP bound intermediate amounts of PhyBt(660). These data suggest that moxGFP has to be present at the GFP-PIF[S]-TCR on the cell surface, in order for PIF[S] to bind to PhyBt(660). The optimized construct, moxGFP-noF-PIF[S]-TCR, will be called GFP-PIF[S]-TCR in the remainder of this article.

In conclusion, through several steps of engineering and optimization we generated the opto-ligand-TCR interaction system (*Figure 1*) based on the red/far-red light-regulated PhyB-PIF pair.

## The GFP-PIF[S]-TCR is switched ON with 660 nm and OFF with 740 nm light

PhyBt(660) bound to cells expressing the GFP-PIF[S]-TCR, whereas PhyBt pre-illuminated with 740 nm light [PhyBt(740)] did not (*Figure 4E,F*). Binding induced TCR signaling, since addition of PhyBt(660), but not PhyBt(740), resulted in a strong calcium influx into the cells similar to a stimulation using an anti-TCR antibody (*Figure 5A,B*). The experiment was done in the dark, since in the absence of any light, the PhyB molecules rest in their state (ON or OFF) for time scales exceeding the duration of the calcium experiments (*Smith et al., 2017*; *Smith et al., 2016*). 660 nm light alone in the absence of PhyBt or GFP-PIF[S]-TCR did not evoke signaling; similarly Jurkat cells not expressing the GFP-PIF[S]-TCR could not be stimulated with PhyBt(660) (*Figure 5—figure supplement 1A,B*). Both experiments show that the light acted through inducing PhyBt binding to GFP-PIF[S]-TCR. Furthermore, as seen with soluble pMHC ligands (*Boniface et al., 1998*; *Cochran et al., 2000*; *Minguet et al., 2007*), PhyB monomers (in contrast to tetramers) could not stimulate calcium influx (*Figure 5—figure supplement 1C*). Lastly, stimulation with bead-coupled PhyBt(660) in the dark resulted in up-

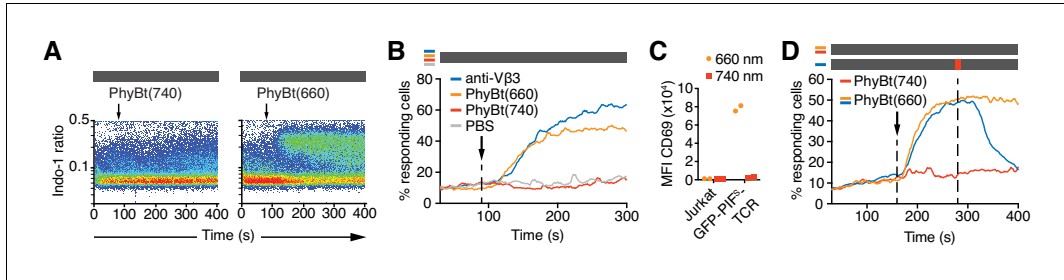

**Figure 5.** The opto-ligand-TCR can be switched ON and OFF. (**A**) GFP-PIF[S]-TCR cells were labeled with Indo-1 and calcium influx measured by flow cytometry. The arrow marks the addition of the stimuli indicated, and the grey rectangle the absence of any light. Results show one experiment of n > 3. (**B**) Calcium influx into GFP-PIF[S]-TCR cells stimulated with anti-Vβ3, PhyBt(660), PhyBt(740) or PBS was measured as in (A). Results show one experiment of n > 3. (**C**) GFP-PIF[S]-TCR Jurkat cells were incubated with PhyBt bound to sepharose beads after a 30 s 660 nm or 740 nm light pulse for 6 hr. Expression of CD69 was quantified by flow cytometry using an APC-labeled anti-CD69 antibody. Data points depict two experiments. (**D**) Calcium influx was measured as in (A). PhyBt (660) induced calcium influx (blue and orange lines). After 2 min a 1 s short pulse of 100% intensity 740 nm light (red break in the grey bar) terminated the calcium response (blue line). Addition of PhyBt(740) did not induce calcium influx (red line). Results show one experiment of n > 3.

DOI: https://doi.org/10.7554/eLife.42475.007

The following figure supplement is available for figure 5:

**Figure supplement 1.** Characterization of the optogenetic PhyBt - GFP-PIF[S]-TCR system.

DOI: https://doi.org/10.7554/eLife.42475.008

regulation of the activation marker CD69 (*Figure 5C*). Together these data show that light-mediated PhyBt-binding to GFP-PIF[S]-TCR induced TCR signaling and T cell activation.

The PhyB-PIF system allows the rapid switching between the ON and OFF states in both directions. When we switched PhyBt from the ON to the OFF state by a 1 s pulse of 740 nm light, we stopped the ongoing calcium response initially evoked by PhyBt(660) (*Figure 5D*), demonstrating that our system is reversible.

## The intensity of continuous 660 nm light determines GFP-PIF[S]-TCR activation

Having established the opto-ligand-TCR system, the second aim of our study was to test the kinetic proofreading (KPR) model.

The KPR model predicts that the half-life of the ligand-TCR interaction determines TCR signaling. Here, we wanted to implement a protocol to control this half-life by light and study the consequences for TCR signaling. To this end, we exploited the property of PhyB that its continuous exposure to 660 nm light triggers both the switch from PhyB OFF to ON and the reverse switch from ON to OFF (*Figure 6A*) as the absorption spectra of both PhyB states partially overlap (*Rockwell et al., 2006*). Thus, each individual PhyB molecule constantly shuttles between the ON and OFF state under 660 nm light, with high 660 nm intensities leading to a faster shuttling rate and thus to shorter binding duration (note that in *Figure 5* continuous light was not used and the PhyB molecules stayed in their ON or OFF state for the duration of the experiment). Accordingly, continuous high intensity (100%) 660 nm light prevented calcium influx when PhyBt(660) was added to the GFP-PIF[S]-TCR cells (*Figure 6B*, orange line). After 390 s the constant 660 nm illumination was stopped, so that the PhyB molecules that were in the ON state at this moment were trapped in this state. This allowed them to bind long enough to the TCR and to induce a strong calcium response (*Figure 6B*). This experiment also demonstrates that the constant high intensity 660 nm illumination did not harm the cells.

The intensity of 660 nm light determines the half-life of both PhyB states and consequently the switch rates between the ON and the OFF state. However, the 80:20 molar ratio of PhyB ON to OFF molecules at photoequilibrium is largely independent of the light intensity (*Figure 6A*) (*Bae and Choi, 2008*; *Smith et al., 2016*). Lowering the 660 nm intensity increases the half-life of PhyB ON without altering its concentration, and hence may allow PhyBt to bind for longer durations to the

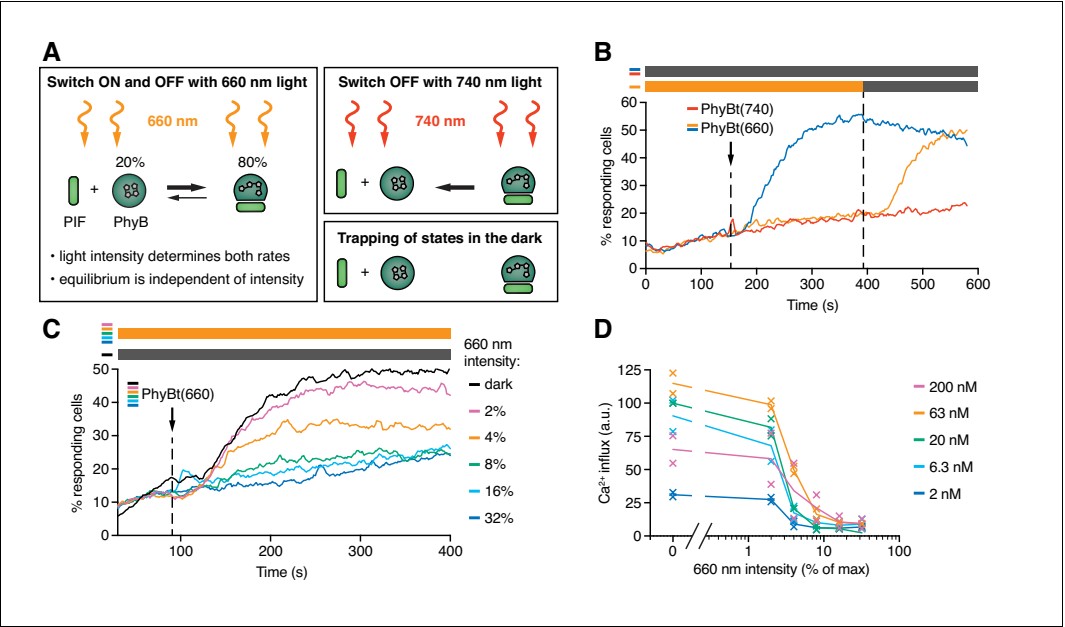

**Figure 6.** The half-life of the ON state of PhyB determines TCR signaling. (**A**) Schematics of the different PhyB conversions under 660 nm and 740 nm light. In the dark the PhyB states do not change in the timescales relevant for this work. (**B**) Calcium influx was measured as in *Figure 5*. GFP-PIF$^S$-TCR cells were constantly illuminated with 100% intensity 660 nm light (orange line). After 150 s PhyBt(660) was added (arrow) and after 390 s the light was switched off. As controls, PhyBt(660) (blue line) or PhyBt(740) (red line) was added to the cells in the dark. The bars represent the illumination procedure during the measurement (grey = dark, orange = 660 nm light). (**C**) 20 nM PhyBt(660) was added (arrow) after 90 s to GFP-PIF$^S$-TCR cells continuously illuminated with 660 nm light of the depicted intensities. Results in (**B**) and (**C**) show one experiment of n > 3. (**D**) Quantification of experiments done as in (**C**) with the indicated PhyBt concentrations. Duplicates are shown with connecting lines going through the mean.

DOI: https://doi.org/10.7554/eLife.42475.009

The following source data and figure supplement are available for figure 6:

**Source data 1.** Calcium influx quantification data at steady-state.
DOI: https://doi.org/10.7554/eLife.42475.011
**Figure supplement 1.** Use of the 660 nm light intensity to tune GFP-PIF$^S$-TCR signaling.
DOI: https://doi.org/10.7554/eLife.42475.010

GFP-PIF$^S$-TCR. Indeed, at 4% and 2% constant 660 nm intensity, calcium influx was evoked (*Figure 6C*). These percentage values refer to the maximum intensity of 100% that was determined by the light source we used. We observed a threshold of the PhyB ON half-life in inducing a calcium response that was largely independent of the PhyBt concentration, a crucial property of TCR ligand discrimination (*McKeithan, 1995*) (*Figure 6D* and *Figure 6—figure supplement 1A*). This threshold half-life was at 3% 660 nm intensity. Thus, we were able to control TCR signaling by changing the intensity of continuous 660 nm light, suggesting that the duration of the ligand-TCR interaction controls calcium signaling.

Next, we tested whether very fast kinetics can terminate an ongoing TCR signal. GFP-PIF$^S$-TCR cells were stimulated with PhyBt(660) in the dark, inducing a strong calcium response (*Figure 6—figure supplement 1B*). During this response the long binding events were changed to fast binding events by illuminating with high intensity continuous 660 nm light (32% and 16%). As expected, the calcium signal was stopped, similar as when using 740 nm light (*Figure 6—figure supplement 1B* and *Figure 5D*). The calcium response was not stopped when low intensity continuous 660 nm light (2% and 4%) was used, where the half-life of binding is still long. The threshold half-life of the PhyBt-GFP-PIF$^S$-TCR interaction to maintain the calcium response was again at 3% 660 nm intensity (*Figure 6—figure supplement 1C*).

In conclusion, we engineered the opto-ligand-TCR system, in which one single ligand-TCR pair explores a wide range of different binding half-lives when changing the intensity of red light and in which other parameters of the interaction remain constant, because we have not mutated the binding interface.

## A mathematical model describing KPR in the opto-ligand-TCR system

Next, we developed a mathematical model and confronted it with the experimental data, to obtain quantitative insight into how the half-life of the PhyB ON-TCR complex determines TCR signaling. The model comprises the PhyB ON-OFF cycle, binding of PhyBt to the TCR, and potentially KPR

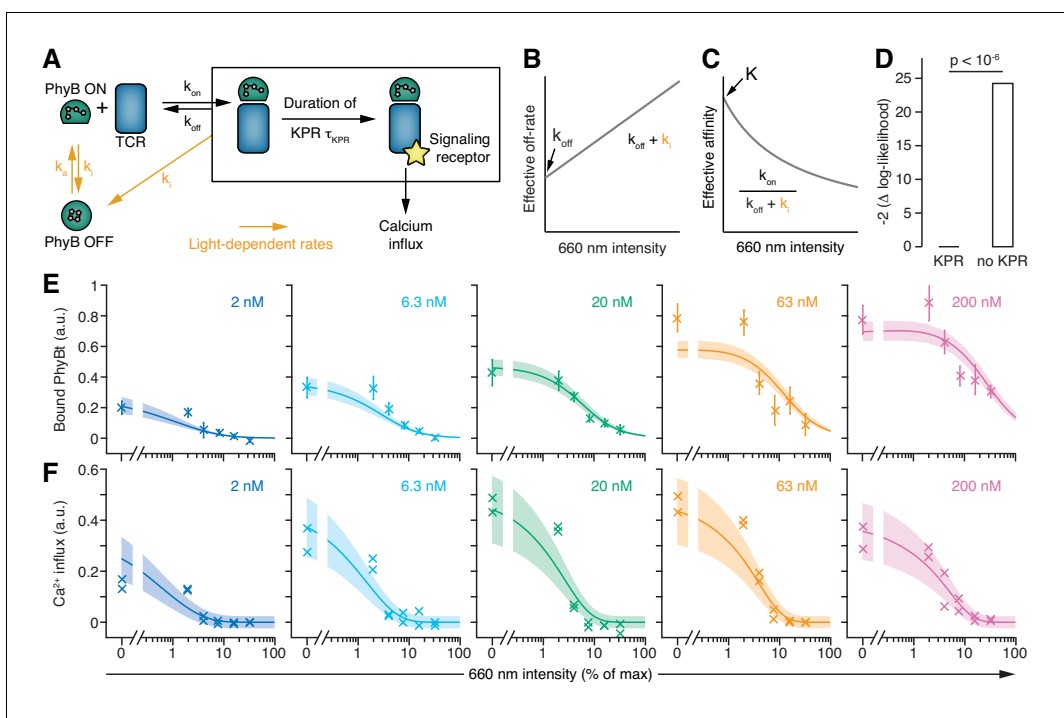

**Figure 7.** T cells exploit a kinetic proofreading mechanism. (**A**) The PhyB ON-OFF cycle, binding of PhyB ON to the TCR and kinetic proofreading (KPR) were combined into one model. (**B**) In this model the effective off-rate is a linear function, and (**C**) the effective affinity is the reciprocal of a linear function, of the 660 nm light intensity. (**D**) A likelihood ratio test (null hypothesis: $\tau_{KPR} = 0$, i.e., no KPR; alternative hypothesis: $\tau_{KPR} > 0$) strongly supports the existence of a KPR mechanism. (**E**) The amount of PhyBt bound to the GFP-PIF$^S$-TCR cells and (**F**) calcium influx at different continuous 660 nm light intensities (from **Figure 6D**) and different PhyBt concentrations are plotted. The line and shaded area represent the fit and the estimated uncertainties of the KPR model. The data points represent the mean ±SEM of 6–9 replicates in (**E**), or individual data points of two experiments in (**F**).
DOI: https://doi.org/10.7554/eLife.42475.012

The following source data and figure supplements are available for figure 7:

**Figure supplement 1.** The duration of the kinetic proofreading mechanism is the principal parameter.
DOI: https://doi.org/10.7554/eLife.42475.013

**Figure supplement 2.** Consecutive setup of the binding model.
DOI: https://doi.org/10.7554/eLife.42475.014

**Figure supplement 3.** Photoconversion rates of PhyB are independent of PIF binding.
DOI: https://doi.org/10.7554/eLife.42475.015

**Figure supplement 4.** Amount of PhyBt bound to GFP-PIF$^S$-TCR inversely correlates with 660 nm light intensity.
DOI: https://doi.org/10.7554/eLife.42475.016

**Figure supplement 4—source data 1.** Quantification data of surface-bound PhyBt.
DOI: https://doi.org/10.7554/eLife.42475.017

**Figure supplement 5.** The data strongly support kinetic proofreading.
DOI: https://doi.org/10.7554/eLife.42475.018

(*Figure 7A*, *Figure 7—figure supplements 1* and *2* and Appendix 2). In the absence of KPR, the activity of each component in the signaling network depends only on the activity of its immediate upstream component(s), making TCR occupancy the ultimate source of ligand discrimination. In contrast, KPR assumes that the first signaling steps at the receptor in addition depend on the half-life of the ligand-TCR complex, while only the more downstream components respond exclusively to the activity of their immediate upstream component(s). We refer to the time required to complete the first half-life-dependent signaling steps as KPR duration or KPR time, $\tau_{KPR}$.

We used a soluble TCR ligand for which - in case of antibodies or pMHC - it was shown that bivalent binding is required to activate the TCR (*Boniface et al., 1998*; *Cochran et al., 2000*; *Kaye and Janeway, 1984*; *Minguet et al., 2007*), and this most likely was also the case for our opto-ligand-TCR system (*Figure 5—figure supplement 1C*). Thus, the KPR duration in our system is the time from bivalent binding to the completion of the biochemical signaling steps (*Figure 7—figure supplement 2*).

KPR requires the bivalently bound PhyBt-TCR complex to exist for at least the KPR time, in order to generate a signal that then leads to a calcium response more downstream (*Altan-Bonnet and Germain, 2005*; *Davis and van der Merwe, 2006*; *Lever et al., 2016*; *McKeithan, 1995*) (*Figure 7A*, bivalent binding is shown in *Figure 7—figure supplement 2*). Thus, the time delay between bivalent ligand binding and calcium influx consists of the KPR duration plus the extra time beyond KPR required for the additional signaling steps until opening of the calcium channels. The half-life of the bivalent PhyBt-TCR complex is determined by the sum of the light-independent off-rate of PhyB ON from the TCR, $k_{off}$, and the light intensity-dependent rate $k_i$ with which PhyB molecules return to the OFF state, detaching from the TCR (*Figure 7B*).

In support of the model (*Figure 7A*), we experimentally demonstrated that the rate of converting PhyB from ON to OFF is the same for free PhyB and PIF-bound PhyB (*Figure 7—figure supplement 3*). These data imply that PhyB molecules also convert to OFF while being bound to PIF and thereby the PhyB-PIF interaction is lost. Hence, the effective off-rate and binding affinity of PhyB ON to the TCR are also light-dependent (*Figure 7B,C*). Taken together, the model predicts that the amount of TCR-bound PhyB decreases with increasing light intensity, which we confirmed experimentally (*Figure 7—figure supplement 4*). Importantly, the change of PhyB ON affinity is a straightforward consequence of the light-controlled PhyB ON half-life (this contrasts with mutated pMHC ligands (*Altan-Bonnet and Germain, 2005*; *Daniels et al., 2006*; *Davis and van der Merwe, 2006*; *Holler and Kranz, 2003*; *Lever et al., 2016*), where affinity changes can be brought about by changes in both on- and off-rates, and potentially other parameters such as orientation of binding (*Adams et al., 2011*)).

## Experimental data and modeling demonstrate that KPR takes place

Although we intended to only change the ligand-TCR half-life with light, we also changed the affinity, due to the intrinsic relationship between off-rate and affinity. Hence, the intensity of 660 nm light regulates both the half-life of PhyB ON and the amount of bound PhyBt. To disentangle the half-life from the amount of ligand-bound TCRs, we asked whether calcium signaling was directly sensitive to the PhyB ON half-life through KPR or solely responded to the level of TCR occupancy with PhyB ON (absence of KPR). We fitted both mathematical models, the one with and the one without KPR, to the PhyBt binding and calcium signaling data together. Only the model with KPR yielded a satisfactory fit, and a likelihood ratio test, with the absence of KPR being the null hypothesis and the presence of KPR being the alternative hypothesis, showed highly significant support for the KPR model ($p < 10^{-6}$, *Figure 7D,E,F* and *Figure 7—figure supplement 5*). Taken together, these findings strongly support the existence of KPR at the TCR.

## The KPR time in Jurkat cells using the opto-ligand-TCR is 8 s

The steady-state data (*Figure 7E,F*) prevented the model to deduce the KPR time $\tau_{KPR}$, yielding only the product $\tau_{KPR} \cdot k_{off}$. To overcome this limitation, we determined the conversion kinetics of PhyB in our experimental system by illuminating PhyBt OFF with short light pulses of 660 nm light and subsequently switching to darkness. This protocol traps the ligands in the ON state, which we quantified through the resulting calcium signal (*Figure 8A* and Appendix 2). The resulting kinetics of switching PhyB to the ON state was highly consistent across different light intensities and PhyBt concentrations

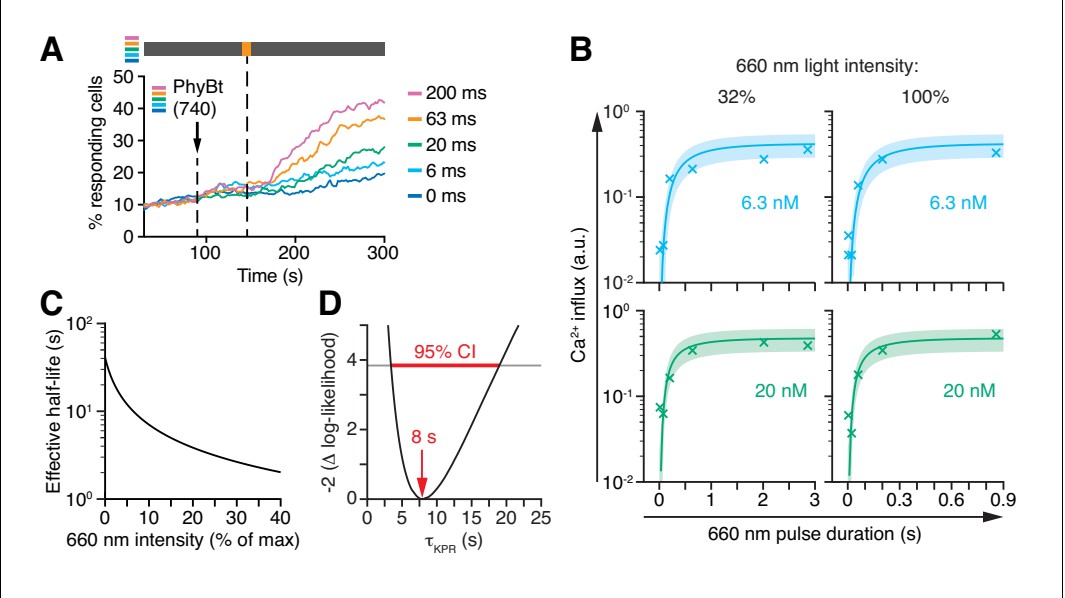

**Figure 8.** Kinetic proofreading at the TCR occurs with a half-life of 8 s. (**A**) 20 nM PhyBt(740) was added to GFP-PIF^S-TCR cells and a 660 nm pulse of 100% intensity was given for the indicated durations. The calcium influx was quantified over time, indicating that longer pulse durations switch more PhyB OFF molecules to the ON state. Stimuli addition is marked by an arrow and the illumination procedure by a bar above the graph (grey = dark, orange = 660 nm). (**B**) Experiment as in (A) were performed using 6.3 nM or 20 nM PhyBt and 32% or 100% intensity 660 nm light. The data is shown together with the fit and estimated SD. Results in (A) and (B) show one experiment of n > 3. (**C**) The estimated half-lives of the PhyB-TCR complex in dependence on the light intensity. (**D**) The profile likelihood of the KPR time shows that the 95% confidence interval (CI) ranges from 3 s to 19 s, while the best-fit value is 8 s.

DOI: https://doi.org/10.7554/eLife.42475.019

The following source data and figure supplement are available for figure 8:

**Source data 1.** Calcium influx quantification data for different 660 nm pulse durations and PhyBt concentrations.
DOI: https://doi.org/10.7554/eLife.42475.021
**Figure supplement 1.** All model parameters are identifiable.
DOI: https://doi.org/10.7554/eLife.42475.020

(*Figure 8B*) and were described well by the mathematical model. Importantly, combining the steady-state data (*Figure 7E,F*) and the kinetic data (*Figure 8B*) was sufficiently informative to identify all five parameters of the model (*Figure 8—figure supplement 1*). Utilizing the kinetic data, we determined the half-life of the PhyB ON-TCR complex, $\ln 2 /(k_{off} + k_i)$, which varied from 40 s to 2 s over the range of light intensities used (*Figure 8C*). We determined the threshold half-life of the bivalent PhyBt-TCR interaction, i.e. the proofreading duration $\tau_{KPR}$, to be 8 s (95% CI: 3 s, 19 s) (*Figure 8D*). Thus, for a threshold half-life of bivalent binding of PhyB to the TCR complex of 8 s, signaling from the active TCR is half-maximal. Furthermore, our results largely exclude the possibility of fast rebinding events, which would have effectively prolonged the half-life of the PhyB ON-TCR complex sensed by a KPR mechanism (*Aleksic et al., 2010*; *Govern et al., 2010*) (Appendix 2).

## Discussion

In this study, we engineered a tailor-made optogenetic system, the opto-ligand-TCR, to control a ligand-receptor interaction by light, allowing us to overcome current experimental limitations. In fact, one single ligand-TCR pair (the PhyBt - GFP-PIF^S-TCR pair) can explore a wide range of different binding half-lives when changing the intensity of 660 nm light. Indeed, our approach exploits the remarkable, but in optogenetics so far unexplored, biophysical property of PhyB that the intensity of 660 nm light determines the half-life of the PhyB ON state (*Bae and Choi, 2008*; *Rockwell et al., 2006*; *Smith et al., 2016*) and thus the half-life of the ligand-TCR interaction. Other parameters of

the interaction remain constant, because the binding interface is always the same under the different light conditions. Together with a mathematical model, our data show that KPR can explain ligand discrimination by T cells.

Furthermore, using the PhyB-PIF pair enables switching ON and OFF ligand binding (short pulse of 660 and 740 nm light, respectively) in less than a second (*Figure 8B*). Importantly, we show that 740 nm light actively disrupts an existing PhyB-PIF interaction, rather than preventing rebinding. Both features, the light-induced switch between both states and the light intensity-dependent change in the binding dynamics, is only found with phytochromes and not with other optogenetic or synthetic systems (*Kolar et al., 2018*; *Smith et al., 2016*).

Previously, light has been used to induce ligand-binding to the TCR. A lysine side chain of the peptide presented by MHC was modified with a light-sensitive caging group (*Huse et al., 2007*). This modified pMHC could not bind to the TCR until a short UV light pulse (microsecond range) removed the caging group. Subsequently, pMHC could bind and stimulate signaling. In contrast to the opto-ligand-TCR system, this approach is not reversible, thus not allowing varying the half-life. Another approach is presented in the accompanying paper by Tischer and Weiner (*Tischer and Weiner, 2019*). It uses a blue-light responsive optogenetic tool, namely the LOVTRAP system (*Wang et al., 2016*). In this case LOV2 binds to a chimeric antigen receptor and the blue light intensity controls the duration of binding. In analogy to our data, they show that the half-life of ligand binding controlled T cell activation.

The opto-ligand-TCR system was not only able to provoke calcium and Erk MAP kinase signaling (not shown), but also led to the stimulation of the T cell as measured by the upregulation of the activation marker CD69. This is in line with systems where other binding domains were fused to the TCR (single chain Fv and DNA, (*Minguet et al., 2007*; *Schamel and Reth, 2012*; *Taylor et al., 2017*), indicating that the TCR can be fully stimulated in synthetic settings and not only by pMHC. This feature is also exploited in chimeric antigen receptors used for cancer immunotherapy (*Lim and June, 2017*; *Sadelain, 2016*).

Our opto-ligand-TCR system allowed us to show that T cells discriminate between ligands due to differences in the ligand-TCR half-lives (*Figure 8*), consistent with KPR models (*Altan-Bonnet and Germain, 2005*; *Davis and van der Merwe, 2006*; *Lever et al., 2016*; *McKeithan, 1995*). Using the identical ligand-TCR pair for the different half-lives excludes differences in binding geometry (*Adams et al., 2011*), forces (*Kim et al., 2009*; *Liu et al., 2014*) or conformational changes (*Gil et al., 2002*) as discriminatory parameters in this setup. Furthermore, we measured total binding, the binding kinetics and the activation readout in the same experimental system. Thus, all parameters for the mathematical model are derived using identical conditions. This is often different when using pMHC and variants thereof as ligands for the TCR: the binding parameters are derived by surface plasmon resonance at 25°C using recombinant parts of the proteins (ectodomains of pMHC and only the immunoglobulin domains of the TCRα and TCRβ subunits) and the activation assays are done with native, membrane or surface bound proteins at 37°C (*Aleksic et al., 2010*; *Dushek et al., 2011*; *Govern et al., 2010*; *Holler and Kranz, 2003*; *Krogsgaard et al., 2003*). Thus, it is often unclear how well these different biological setups can be compared and compiled into one model.

Besides our and other studies on the correlation of binding parameters with the biological activity of the ligands, differential CD3ζ phosphorylation is another hint for KPR. CD3ζ is a signaling subunit of the TCR that can be partially or fully phosphorylated. Low affinity pMHC, which are non-stimulatory, lead to partial phosphorylation, whereas high affinity pMHC, which are stimulatory, lead to full phosphorylation of CD3ζ (*Madrenas et al., 1995*; *van Oers et al., 1993*). This is consistent with the idea that the low affinity ligands only bind shortly, not allowing all phosphorylation steps to be completed and high affinity ligands bind long enough to complete all phosphorylations. Indeed, increasing the concentration of the low affinity binders did not lead to full CD3ζ phosphorylation (*Madrenas et al., 1997*), being consistent with KPR.

Interestingly, changing the half-life of PhyB ON and thus the lifetime of the ligand-TCR interaction also altered the amount of bound receptors, and with the help of the mathematical model we could show that the half-life was the decisive parameter for the magnitude of T cell stimulation as measured by calcium influx. We calculated the threshold half-life above which TCR stimulation occurs, i. e., the KPR duration, to be 8 s. For soluble TCR ligands, as we have used here, it has been shown that bivalent binding is required to trigger the TCR (*Boniface et al., 1998*; *Cochran et al., 2000*;

*Kaye and Janeway, 1984*), possibly due to both a lack of clustering and stabilization of conformational changes of the TCR (*Minguet et al., 2007*; *Schamel et al., 2017*; *Swamy et al., 2016*). Indeed, also in our case PhyB monomers did not activate the TCR whereas PhyB tetramers (PhyBt) did. Thus, the PhyBt ligands needed to bind for at least 8 s bivalently, in order to stimulate calcium influx that itself occurred later. Of note, the KPR duration is not identical to the time delay between ligand binding and calcium influx or other downstream events (*Figure 9*). A time delay is a prerequisite for KPR, but does not necessarily indicate that KPR takes place.

In line with our 8 s KPR time, the accompanying paper by Tischer and Weiner found a KPR time of approximately 7 s (*Tischer and Weiner, 2019*). This study also used Jurkat cells, but a different optogenetic system, a different activation readout and a chimeric antigen receptor instead of a TCR. Thus, independent of the readout and exact design of the optogenetic system Jurkat cells have a TCR/-chimeric antigen receptor based KPR time of 7–8 s. Most other studies have calculated a KPR time of between 1–5 s (*Aleksic et al., 2010*; *Altan-Bonnet and Germain, 2005*; *Daniels et al., 2006*; *Govern et al., 2010*; *Holler and Kranz, 2003*; *Kersh et al., 1998*) and the time delay between ligand binding and calcium influx was 7 s in one study (*Huse et al., 2007*). In contrast to those studies, our and the Tischer/Weiner systems lack the co-receptor CD4 and CD8 that have been shown to increase the speed of signaling, most likely by efficiently recruiting the kinase Lck to the TCR (*Artyomov et al., 2010*; *Holler and Kranz, 2003*; *Veillette et al., 1988*). Differences in the cellular background (primary murine T cells versus the human T cell line Jurkat) might also contribute to differences in the KPR time, e.g., if the concentration of kinases or phosphatases was different (*Altan-Bonnet and Germain, 2005*).

The half-life of the interaction is a population average over many binding events. Thus, it might be that individual binding events longer than the threshold half-life (8 s in our case) are the ones that triggered T cell activation, as suggested recently (*Lin et al., 2019*). The opto-ligand-TCR system is well suited to precisely control the exact binding time (and not the average half-life) by using 740 nm light to break the ligand-TCR interaction.

An aspect to consider in KPR is a potential contribution of fast rebinding of ligands to TCRs (*Aleksic et al., 2010*; *Govern et al., 2010*). When the on-rate of multivalent binding of the pMHC-TCR interaction is sufficiently fast, dissociated TCRs are rebound before KPR modifications are removed (*Aleksic et al., 2010*; *Govern et al., 2010*), effectively prolonging the half-life of the TCR-ligand interaction. However, the on-rate in our opto-ligand-TCR system seems to be too slow to significantly contribute to this effect (Appendix 2).

Our approach, including the designed PIF$^S$ mutant, could be a blueprint to study other ligand-receptor pairs and to understand how the kinetics of protein-protein interactions governs the activity

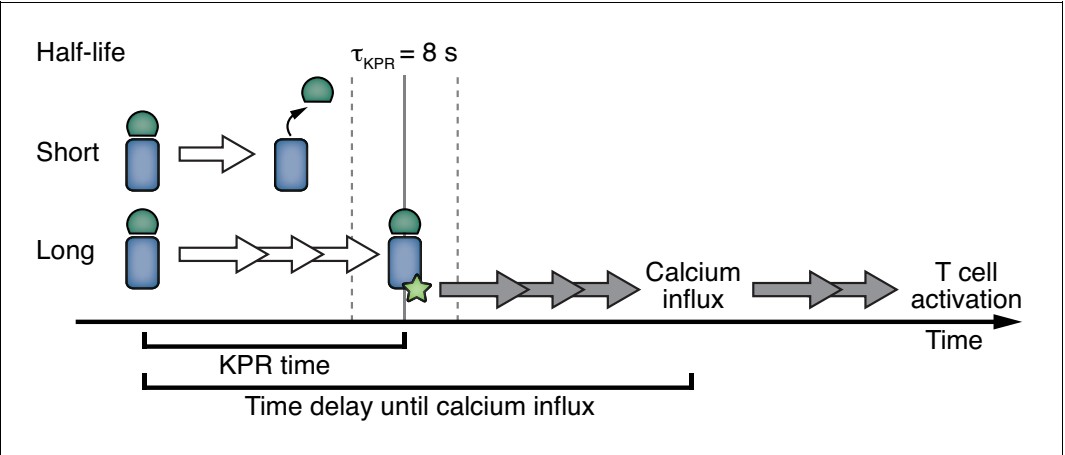

**Figure 9.** Kinetic proofreading determines T cell activation. Ligands that bind shorter than the KPR time of 8 s (half-life of binding) fail to induce efficient TCR signaling. Ligands that bind longer allow the completion of several biochemical steps (white arrows) that result in an activatory signal by the TCR. This signal provokes further signaling (grey arrows) that ultimately leads to T cell activation.
DOI: https://doi.org/10.7554/eLife.42475.022

of these binding events in diverse biological systems. Further, the opto-ligand-receptor approach is also well suited to locally induce signaling by focusing the light beam to the region of interest.

# Materials and methods

## Key resources table

| Reagent type (species) or resource | Designation | Source or reference | Identifiers | Additional information |
|---|---|---|---|---|
| Genetic reagent (Aequorea victoria) | moxGFP | PMID: 26158227 | | Erik Snapp (Albert Einstein College of Medicine), Addgene plasmid # 68070 |
| Genetic reagent (Arabidopsis thaliana) | PIF6 | PMID: 29603429 | | |
| Genetic reagent (Homo sapiens) | HA1.7 TCRβ | PMID: 17188005 | | |
| Genetic reagent (Mus musculus) | scFv | PMID: 17188005 | | |
| Cell line (Homo sapiens) | Jurkat | PMID: 15057788 | | Arthur Weiss (HHMI, UCSF) |
| Cell line (Homo sapiens) | Jurkat scFv-TCRβ | this paper | | Jurkat expressing scFv-TCRβ |
| Cell line (Homo sapiens) | Jurkat PIF(wt)-TCRβ | this paper | | Jurkat expressing PIF(wt)-TCRβ |
| Cell line (Homo sapiens) | Jurkat PIF(Q)-TCRβ | this paper | | Jurkat expressing PIF(Q)-TCRβ |
| Cell line (Homo sapiens) | Jurkat PIF(A)-TCRβ | this paper | | Jurkat expressing PIF(A)-TCRβ |
| Cell line (Homo sapiens) | Jurkat PIF(SS)-TCRβ | this paper | | Jurkat expressing PIF(SS)-TCRβ |
| Cell line (Homo sapiens) | Jurkat PIF(SSQ)-TCRβ | this paper | | Jurkat expressing PIF(SSQ)-TCRβ |
| Cell line (Homo sapiens) | Jurkat PIF(SSA)-TCRβ; Jurkat PIF$^S$-TCRβ | this paper | | Jurkat expressing PIF(SSA)-TCRβ |
| Cell line (Homo sapiens) | Jurkat GFP-F1-PIF$^S$-TCRβ | this paper | | Jurkat expressing GFP-F1-PIF$^S$-TCRβ |
| Cell line (Homo sapiens) | Jurkat GFP-F2-PIF$^S$-TCRβ | this paper | | Jurkat expressing GFP-F2-PIF$^S$-TCRβ |
| Cell line (Homo sapiens) | Jurkat GFP-F3-PIF$^S$-TCRβ | this paper | | Jurkat expressing GFP-F3-PIF$^S$-TCRβ |
| Cell line (H. sapiens) | Jurkat GFP-noF-PIF$^S$-TCRβ; Jurkat GFP-PIF$^S$-TCRβ | this paper | | Jurkat expressing GFP-noF-PIF$^S$-TCRβ |
| Antibody | anti-Vβ3 Jovi3 | Ancell Cat# 102-020 | - | 5 µg/ml |
| Antibody | biotin-conjugated anti-GFP | Rockland Cat# 600-106-215 | RRID:AB_218204 | 5 µg/ml |
| Antibody | APC-conjugated anti-CD69 | Thermo Fisher Cat# MHCD6905 | RRID:AB_10372807 | 1:200 |
| Antibody | APC-conjugated anti-mouse | SouthernBiotech Cat# 1031-11L | - | 1:200 |
| Recombinant DNA reagent | PhyB$_{1-651}$-Aviag-His$_6$; pMH17 | PMID: 27884151 | | |
| Recombinant DNA reagent | p171 | PMID: 18832155 | | Lars-Oliver Essen (University Marburg) |
| Recombinant DNA reagent | PIF(wt)-TCRβ; pOSY015 | this paper | | see Table S1 |
| Recombinant DNA reagent | PIF(Q)-TCRβ; pOSY016 | this paper | | see Table S1 |

*Continued on next page*

*Continued*

| Reagent type (species) or resource | Designation | Source or reference | Identifiers | Additional information |
|---|---|---|---|---|
| Recombinant DNA reagent | PIF(A)-TCRβ; pOSY017 | this paper | | see Table S1 |
| Recombinant DNA reagent | scFv-TCRβ; pOSY019 | this paper | | see Table S1 |
| Recombinant DNA reagent | PIF(SS)-TCRβ; pOSY026 | this paper | | see Table S1 |
| Recombinant DNA reagent | PIF(SSQ)-TCRβ; pOSY027 | this paper | | see Table S1 |
| Recombinant DNA reagent | PIF(SSA)-TCRβ; PIF$^S$-TCRβ; pOSY028 | this paper | | see Table S1 |
| Recombinant DNA reagent | MBP-PIF(wt); pOSY061 | this paper | | see Table S1 |
| Recombinant DNA reagent | MBP-PIF(Q); pOSY062 | this paper | | see Table S1 |
| Recombinant DNA reagent | MBP-PIF(A); pOSY063 | this paper | | see Table S1 |
| Recombinant DNA reagent | MBP-PIF(SS); pOSY064 | this paper | | see Table S1 |
| Recombinant DNA reagent | MBP-PIF(SSQ); pOSY065 | this paper | | see Table S1 |
| Recombinant DNA reagent | MBP-PIF(SSA); pOSY066 | this paper | | see Table S1 |
| Recombinant DNA reagent | GFP-F1-PIF$^S$-TCRβ; pOSY073 | this paper | | see Table S1 |
| Recombinant DNA reagent | GFP-F2-PIF$^S$-TCRβ; pOSY074 | this paper | | see Table S1 |
| Recombinant DNA reagent | GFP-F3-PIF$^S$-TCRβ; pOSY075 | this paper | | see Table S1 |
| Recombinant DNA reagent | GFP-noF-PIF$^S$-TCRβ; GFP-PIF$^S$-TCRβ; pOSY076 | this paper | | see Table S1 |
| Peptide, recombinant protein | PE-conjugated streptavidin | Thermo Fisher Cat# S866 | - | |
| Peptide, recombinant protein | DyLight650-conjugated streptavidin | Thermo Fisher Cat# 84547 | - | |

## Molecular cloning

All plasmids generated in this study were created using standard molecular cloning techniques like polymerase chain reaction, restriction enzyme digestion and ligation or Gibson assembly (*Gibson et al., 2009*). The plasmids are listed in Table S1 together with the corresponding coded protein, a brief description of the used components and the cloning strategy. The primers used as described in *Supplementary file 1* are summarized in *Supplementary file 2*. Plasmid maps and nucleotide sequences in GeneBank format are available as supplementary information. Plasmid maps were generated with Geneious 6.1.8 (https://www.geneious.com). The integrity of each plasmid was verified by restriction enzyme digestion and Sanger sequencing. The plasmid containing moxGFP was a gift from Erik Snapp (Addgene plasmid # 68070) (*Costantini et al., 2015*).

## Protein production and purification

The production of PhyB$_{1-651}$-Aviag-His$_6$ was performed similarly as described before (*Smith et al., 2016*). Briefly, the PhyB-coding plasmid pMH17 was co-transformed into *E. coli* BL21(DE3) with plasmid p171 (*Rohmer et al., 2008*), which codes for the *Synechocystis* enzymes heme oxygenase and

phycocyanobilin synthase, necessary for phycocyanobilin production. Co-transformed cells were selected with 100 µg/ml ampicillin and 40 µg/ml kanamycin. Bacterial cultures were grown at 30°C in lysogeny broth until $OD_{600}$ reached 0.6, then expression was induced with 1 mM isopropyl β-D-1-thiogalactopyranoside (IPTG) and 0.4% (w/v) arabinose in the presence of 50 µM biotin. Protein production was sustained for 20 hr at 18°C in the dark. Following centrifugation of the bacteria for 8 min at 6500 $g$, the cells were resuspended in lysis buffer (50 mM HEPES, 500 mM NaCl, 5% glycerol, 0.5 mM TCEP, 20 mM imidazole, pH 7.4) and disrupted using a French Press (APV 2000, APV Manufacturing) at 1,000 bar. The lysate was cleared from debris by centrifuging twice at 30,000 $g$ at 4°C for 30 min. The cleared lysate was loaded onto a Ni-NTA Superflow cartridge (Qiagen) using an Äkta Explorer chromatography system (GE Healthcare). After washing with 30 column volumes lysis buffer, purified $PhyB_{1-651}$-Avitag-$His_6$ was eluted with 10 column volumes elution buffer (50 mM HEPES, 500 mM NaCl, 5% glycerol, 0.5 mM TCEP, 500 mM imidazole, pH 7.4). The eluate fractions containing the purified proteins were pooled and the buffer was exchanged to PBS (phosphate-buffered saline, Sigma-Aldrich) containing 0.5 mM TCEP and 10% glycerol using a HiPrep 26/10 desalting column (GE Healthcare).

The expression and purification of the different $MBP-PIF6_{1-100}$ proteins was performed analogous to PhyB, with the difference that the plasmids pOSY061 until pOSY066 were transformed individually without p171, protein expression was induced using only IPTG and no biotin was added to the medium.

PhyB tetramers (PhyBt) were formed by mixing Ni-NTA column-purified $PhyB_{1-651}$-Avitag-$His_6$ in a 10:1 molar ratio with PE- or DyLight650-conjugated streptavidin (Thermo Fisher) and incubating the mixture for 2 hr at room temperature in the dark. The formed PhyB tetramers were separated from the excess of PhyB monomers by size-exclusion chromatography on a HiLoad Superdex 200 pg column (GE Healthcare) using PBS with 0.5 mM TCEP as running buffer.

## Analytical size-exclusion chromatography

To test the interaction of PhyB and MBP-PIF, PhyB was illuminated with saturating amounts of 660 or 740 nm light and MBP-PIF was added as depicted. Following incubation for 1 hr at room temperature in the dark, the samples were separated by size-exclusion chromatography on a Superdex 200 10/300 GL column (GE Healthcare) using PBS with 0.5 mM TCEP as running buffer.

## Cell line generation and cultivation

Jurkat E6.1 and derived cell lines were cultivated in RPMI 1640 medium supplemented with 10% fetal bovine serum (FBS), 2 mM L-glutamine, 10 mM HEPES, 100 U/ml penicillin and 100 µg/ml streptomycin (all Thermo Fisher) at 37°C in a humidified atmosphere of 5% $CO_2$. HEK 293 T cells were cultured in DMEM (Thermo Fisher) supplemented as the RPMI medium at 37°C in a humidified atmosphere of 7.5% $CO_2$.

For the generation of Jurkat-based cell lines stably expressing the chimeric TCRβ chains, we used lentiviral transduction as described earlier (*Dopfer et al., 2014*). Briefly, HEK 293 T cells were transfected with the lentiviral packaging plasmid pCMV dR8.74, the envelope plasmid pMD2 vsvG (both kind gifts from Didier Trono) and the respective transfer plasmid by calcium phosphate precipitation. 6 hr post-transfection the medium was replaced and lentiviral particles were produced by the HEK 293 T cells for 48 hr. Lentiviral particle-containing HEK 293T supernatant was harvested, filtered through a 0.45 µm syringe filter and concentrated by overnight centrifugation at 3,000 $g$ at 4°C through a 20% (w/v in PBS) sucrose cushion. After discarding the supernatant, the viral particles were resuspended in medium using 1/100[th] of the harvested volume. Jurkat cells were transduced with different dilutions of concentrated lentiviral particles and 48 hr after transduction, surface expression and cell viability were analyzed by flow cytometry.

The identity of the Jurkat cells was confirmed by the binding to the antibody C305 that only binds to the TCR expressed on Jurkat cells (*Weiss and Stobo, 1984*). The identity of the HEK 293 T cells was not confirmed. All cells were routinely tested for mycoplasma and devoid of contamination.

## Cell surface staining for flow cytometry

Cells were stained for surface proteins according to standard protocols. Briefly, cells were washed once with washing buffer (PBS supplemented with 1% FBS), then incubated for 30 min at 4°C in a

diluted solution of the labeling antibody as depicted in the key resources table. Finally, the cells were washed twice as before and analyzed on a MACSQuant X flow cytometer (Miltenyi). The labeling reagents used in this study were anti-Vβ3 Jovi3 (Ancell), biotin-conjugated anti-GFP (Rockland Immunochemicals), APC-conjugated anti-CD69 (Thermo Fisher), APC-conjugated anti-mouse (Thermo Fisher) and PE-conjugated streptavidin (Thermo Fisher).

### Light-dependent PhyBt binding to the cell surface

Binding of PhyBt to the different cells lines was performed analogous to the cell surface staining with antibodies, but instead of labeled antibodies 100 nM pre-illuminated Phycoerythrin (PE)-labeled PhyBt(660) or PhyBt(740) were added to the cells and incubated for 30 min at 4°C in the dark. Subsequent washing steps and the measurement at the flow cytometer were executed under green light.

To evaluate the amount of surface bound PhyBt under constant illumination with varying intensities of 660 nm light, different concentrations of PhyBt(660) were added to GFP-PIF$^S$-TCR cells under illumination conditions as depicted and incubated for 90 s at 37°C. Subsequently, the cells were transferred to a ten-fold excess of ice-cold washing buffer, immediately centrifuged for 10 s under green light and the supernatant aspirated. After a second washing step, surface-bound PhyBt was quantified by flow cytometry. Unspecific binding was accounted for and subtracted from each sample by adding ten-fold diluted amounts of PhyBt(660) to control samples that were treated with washing buffer during the 90 s incubation step.

### Calcium influx measurement

Five million cells were centrifuged for 5 min at 300 $g$ and the medium was discarded. The cell pellet was resuspended in 1 ml stimulation medium (RPMI 1640 medium supplemented with 1% FBS, 2 mM L-glutamine, 10 mM HEPES, 100 U/ml penicillin and 100 µg/ml streptomycin) with 0.1% (v/v) pluronic F-127 and 4 µM Indo-1 AM (all Thermo Fisher) and incubated in the dark for 30 min at 37°C. The stained cells were washed and kept on ice in the dark until the measurement. For calcium influx, cells were diluted 1:20 with pre-warmed stimulation medium and maintained at 37°C during the event collection on a MACSQuant X flow cytometer. After fluorescence baseline acquisition, stimuli were added or activated by illumination as depicted. If not indicated otherwise PhyBt were added to a final concentration of 20 nM.

For the graphs showing the percent of responding cells, the events above the 90[th] percentile during baseline acquisition were quantified using FlowJo 9 (FlowJo LLC). To calculate the calcium influx values (a.u.), average Indo-1 ratio values after stimuli addition (250–400 s) minus baseline values (30–60 s) were normalized for each experiment using an internal control of 20 nM PhyBt(660) in the dark.

### CD69 upregulation

200,000 Jurkat or GFP-PIF$^S$-TCR cells were seeded per well in a 96-well flat-bottom plate in 100 µl stimulation medium and incubated for 1 hr in the cultivating incubator. Meanwhile, streptavidin sepharose beads (GE Healthcare) were washed with PBS and then incubated with 5 µg purified PhyB per µl beads (diluted in PBS) at 37°C for 30 min. The beads were washed twice with PBS and resuspended in stimulation medium at 2 µl beads per 100 µl medium. The diluted beads were illuminated as described, 100 µl bead suspension added per well to the cells and the cells stimulated for 6 hr in the incubator. Following the incubation, surface expression of CD69 was analyzed by flow cytometry as described above.

### Determination of PhyB conversion rates

50 µg purified PhyB(660) or PhyB(740) was mixed with a 6-fold excess of MBP-PIF(wt) or an equal volume of buffer (PBS with 0.5 mM TCEP) and incubated for 60 min at room temperature. Each protein mixture was transferred to a quartz cuvette, a blank measurement was taken and under constant illumination with 70 µmol m$^{-2}$ s$^{-1}$ 660 or 740 nm light difference absorbance spectra were acquired every 10 s using a HR4000 spectrometer in combination with a DT-Mini-2-GS light source (Ocean Optics). We quantified the conformational change of PhyB by subtracting the minimum absorbance value from the maximum value and plotted this ΔΔA value against the time of illumination (not shown). From the resulting curves, we calculated the photoconversion rates by first order association

kinetics nonlinear regression using the software Prism 6 (GraphPad Software). Differences in the conversion rates with or without MBP-PIF were tested by two-way ANOVA using Prism 6.

## Illumination devices

For the different experiments performed in this study, we used two types of illumination devices. One device was built as a closed box with an array of red (Osram, LH W5AM, Mouser Electronics) and far-red (LZ4-00R308, LED Engin) light-emitting diodes (LEDs) at the top, resulting in a planar light source. Ventilated openings in the box in combination with light traps allowed gas exchange for the use of the device in an incubator. This illumination box was used for all pre-illumination steps, the CD69 upregulation experiments and PhyB conversion rate measurements.

The second device was built together with Opto Biolabs as a cylinder enclosing a reaction tube in the center. Surrounding the reaction tube, is a water-filled space, which is connected to a 37°C water bath to keep a physiological temperature. Further outside we placed rings of red (Super Bright Red, Kingbright Electronic Europe) and far-red (LED740 series, Roithner Lasertechnik) LEDs, pointing towards the reaction tube. An opaque outmost cylinder shields the sample from external light. The cylindrical illumination device was used for all calcium experiments and experiments under constant 660 nm illumination in combination with a MACSQuant X flow cytometer.

## Repetition of experiments and data presentation

In this study, all graphs derived from data of multiple experiments depict individual data points for less than three replicates and average values for three or more replicates. The uncertainties of these experiments are shown by the standard error of the mean (SEM). For graphs displaying representative experiments, 'n' in the legend defines the number of independent experiments that the depicted results were done.

## Acknowledgements

We thank Susana Minguet for discussions and reading of the manuscript, Susan Lauw and Johannes Kaiser from the BIOSS toolbox for running of the robotic platform (INST 39/899–1 FUGG) to conduct the calcium experiments, and Opto Biolabs for their customized illumination devices. This work was funded by the Deutsche Forschungsgemeinschaft (DFG) under Germany's Excellence Strategy through EXC294 (BIOSS - Center for Biological Signalling Studies), and EXC2189 (CIBSS – Centre for Integrative Biological Signalling Studies, Project ID 390939984) and GSC-4 (Spemann Graduate School, OSY and MH). TH is a member of CellNetworks.

## Additional information

### Funding

| Funder | Grant reference number | Author |
|---|---|---|
| Deutsche Forschungsgemeinschaft | EXC2189 | O Sascha Yousefi<br>Maximilian Hörner |
| Deutsche Forschungsgemeinschaft | GSC-4 | Matias D Zurbriggen<br>Wilfried Weber |
| Deutsche Forschungsgemeinschaft | EXC294 | Thomas Höfer |
| Deutsche Forschungsgemeinschaft | EXC81 | Wolfgang WA Schamel |
| Deutsche Forschungsgemeinschaft | INST 39/899-1 FUGG | Wolfgang WA Schamel |

The funders had no role in study design, data collection and interpretation, or the decision to submit the work for publication.

### Author contributions

O Sascha Yousefi, Conceptualization, Formal analysis, Validation, Investigation, Visualization, Methodology, Writing—original draft, Writing—review and editing; Matthias Günther, Software, Formal analysis, Visualization, Methodology, Writing—original draft, Writing—review and editing; Maximilian Hörner, Resources, Investigation, Writing—review and editing; Julia Chalupsky, Maximilian Wess, Simon M Brandl, Investigation; Robert W Smith, Christian Fleck, Tim Kunkel, Methodology; Matias D Zurbriggen, Conceptualization, Methodology; Thomas Höfer, Conceptualization, Supervision, Funding acquisition, Methodology, Writing—review and editing; Wilfried Weber, Conceptualization, Resources, Methodology; Wolfgang WA Schamel, Conceptualization, Supervision, Funding acquisition, Methodology, Writing—original draft, Project administration, Writing—review and editing

### Author ORCIDs

O Sascha Yousefi (iD) https://orcid.org/0000-0001-5304-729X
Matthias Günther (iD) http://orcid.org/0000-0001-8077-8194
Maximilian Hörner (iD) http://orcid.org/0000-0003-1743-9581
Robert W Smith (iD) https://orcid.org/0000-0001-9657-7477
Wolfgang WA Schamel (iD) http://orcid.org/0000-0003-4496-3100

### Decision letter and Author response

Decision letter https://doi.org/10.7554/eLife.42475.063
Author response https://doi.org/10.7554/eLife.42475.064

## Additional files

### Supplementary files

• Supplementary file 1. List of plasmids created in this study. Plasmids generated in this study were listed next to the protein each plasmid codes for and a brief description. Detailed descriptions of the cloning strategies are available upon request.
DOI: https://doi.org/10.7554/eLife.42475.023

• Supplementary file 2. Summary of primers.
DOI: https://doi.org/10.7554/eLife.42475.024

• Supplementary file 3. Plasmid map pOSY015.
DOI: https://doi.org/10.7554/eLife.42475.025

• Supplementary file 4. Plasmid pOSY015 sequence.
DOI: https://doi.org/10.7554/eLife.42475.026

• Supplementary file 5. Plasmid pOSY016 sequence.
DOI: https://doi.org/10.7554/eLife.42475.027

• Supplementary file 6. Plasmid map pOSY016.
DOI: https://doi.org/10.7554/eLife.42475.028

• Supplementary file 7. Plasmid pOSY017 sequence.
DOI: https://doi.org/10.7554/eLife.42475.029

• Supplementary file 8. Plasmid map pOSY017.
DOI: https://doi.org/10.7554/eLife.42475.030

• Supplementary file 9. Plasmid pOSY019 sequence.
DOI: https://doi.org/10.7554/eLife.42475.031

• Supplementary file 10. Plasmid map pOSY019.
DOI: https://doi.org/10.7554/eLife.42475.032

• Supplementary file 11. Plasmid pOSY026 sequence.
DOI: https://doi.org/10.7554/eLife.42475.033

• Supplementary file 12. Plasmid map pOSY026.
DOI: https://doi.org/10.7554/eLife.42475.034

• Supplementary file 13. Plasmid pOSY027 sequence.

DOI: https://doi.org/10.7554/eLife.42475.035

• Supplementary file 14. Plasmid map pOSY027.
DOI: https://doi.org/10.7554/eLife.42475.036

• Supplementary file 15. Plasmid pOSY028 sequence.
DOI: https://doi.org/10.7554/eLife.42475.037

• Supplementary file 16. Plasmid map pOSY028.
DOI: https://doi.org/10.7554/eLife.42475.038

• Supplementary file 17. Plasmid pOSY061 sequence.
DOI: https://doi.org/10.7554/eLife.42475.039

• Supplementary file 18. Plasmid map pOSY061.
DOI: https://doi.org/10.7554/eLife.42475.040

• Supplementary file 19. Plasmid pOSY062 sequence.
DOI: https://doi.org/10.7554/eLife.42475.041

• Supplementary file 20. Plasmid map pOSY062.
DOI: https://doi.org/10.7554/eLife.42475.042

• Supplementary file 21. Plasmid pOSY063 sequence.
DOI: https://doi.org/10.7554/eLife.42475.043

• Supplementary file 22. Plasmid map pOSY063.
DOI: https://doi.org/10.7554/eLife.42475.044

• Supplementary file 23. Plasmid pOSY064 sequence.
DOI: https://doi.org/10.7554/eLife.42475.045

• Supplementary file 24. Plasmid map pOSY064.
DOI: https://doi.org/10.7554/eLife.42475.046

• Supplementary file 25. Plasmid pOSY065 sequence.
DOI: https://doi.org/10.7554/eLife.42475.047

• Supplementary file 26. Plasmid map pOSY065.
DOI: https://doi.org/10.7554/eLife.42475.048

• Supplementary file 27. Plasmid pOSY066 sequence.
DOI: https://doi.org/10.7554/eLife.42475.049

• Supplementary file 28. Plasmid map pOSY066.
DOI: https://doi.org/10.7554/eLife.42475.050

• Supplementary file 29. Plasmid pOSY073 sequence.
DOI: https://doi.org/10.7554/eLife.42475.051

• Supplementary file 30. Plasmid map pOSY073.
DOI: https://doi.org/10.7554/eLife.42475.052

• Supplementary file 31. Plasmid pOSY074 sequence.
DOI: https://doi.org/10.7554/eLife.42475.053

• Supplementary file 32. Plasmid map pOSY074.
DOI: https://doi.org/10.7554/eLife.42475.054

• Supplementary file 33. Plasmid pOSY075 sequence.
DOI: https://doi.org/10.7554/eLife.42475.055

• Supplementary file 34. Plasmid map pOSY075.
DOI: https://doi.org/10.7554/eLife.42475.056

• Supplementary file 35. Plasmid pOSY076 sequence.
DOI: https://doi.org/10.7554/eLife.42475.057

• Supplementary file 36. Plasmid map pOSY076.
DOI: https://doi.org/10.7554/eLife.42475.058

• Transparent reporting form
DOI: https://doi.org/10.7554/eLife.42475.059

### Data availability

All data that were analyzed with the mathematical model are provided in source data files.

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

## Appendix 1

DOI: https://doi.org/10.7554/eLife.42475.060

# Glossary

| | |
|---|---|
| PhyB | Phytochrome B (photoreceptor of *Arabidopsis thaliana*), here only the first 651 amino acids are used, since this part is sufficient to bind PIF under 660 nm, but not under 740 nm light. |
| PIF | PhyB-interacting factor, here the first 100 aa of *A. thaliana* PIF6 are used, since they are sufficient to bind to PhyB. |
| PIF$^S$ | A mutant version of functional PIF developed in this study, that can move through the secretory pathway to the cell surface and remain in a functional conformation, i.e., it is still capable of binding PhyB. |
| PhyBt | Fluorescently labeled streptavidin-based tetramers of PhyB. |
| PhyBt(660) | PhyBt pre-illuminated with 660 nm light, so that equilibrium is reached, in which 80% of the PhyB molecules are in their ON state |
| PhyBt(740) | PhyBt pre-illuminated with 740 nm light, so that all PhyB molecules are in the OFF state |
| PhyB ON | PhyB in the active conformation allowing binding to PIF with nanomolar affinity (**Figure 1**). |
| PhyB OFF | PhyB in the inactive conformation, in which they cannot bind to PIF (**Figure 1**). |
| $\tau_{KPR}$ | The KPR time or KPR duration. The half-life of the bivalent binding of the soluble ligand to the TCR needs to be at least this time to induce downstream, activatory signaling. |
| moxGFP | A GFP mutant (S30R, Y29N, C48S, F64L, S65T, C70S, Q80R, F99S, N105T, Y145F, M153T, V163A, I171V, A206K) that was optimized for the folding in the endoplasmic reticulum (ox) and that is expressed as a monomer (m). |
| TCR | T cell receptor. The receptor on T cells controlling T cell activation. It consists of the TCR$\alpha$, TCR$\beta$, CD3$\gamma$, CD3$\delta$, CD3$\varepsilon$ and $\zeta$ chains. |
| PIF$^S$-TCR | A preliminary non-functional TCR chimera, in which PIF$^S$ is fused to the ectodomain of the TCR$\beta$ subunit. |
| pMHC | Peptide bound to Major Histocompatibility Complex constitutes the ligand for the TCR. |
| GFP-PIF$^S$-TCR | Our engineered TCR, in which the moxGFP and PIF$^S$ are fused to the ectodomain of the TCR$\beta$ subunit. |
| GFP-PIF$^S$-TCR cells | Jurkat E6.1 T cells expressing the GFP-PIF$^S$-TCR. |

## Appendix 2

DOI: https://doi.org/10.7554/eLife.42475.060

# Mathematical model of TCR activation by the optogenetic ligand

### Dynamics of phytochrome B conversion

Phytochrome B (PhyB) exists in two conformations that can be reversibly interconverted by light with intensity-dependent rates $k_a$ and $k_i$ (**Figure 7—figure supplement 2A**). The two conformations, which are referred to as PhyB ON respectively PhyB OFF, are distinguished by their ability to interact with PhyB interacting factor (PIF), as described in the next section. The rates of conformational conversion depend on the wavelength $\lambda$ and the intensity $I$ of the light. While the dependence on $\lambda$ is very complicated, the rates depend linearly on the intensity (**Smith et al., 2016**),

$$k_{\mathrm{a}}(\lambda, I) = c_a(\lambda) \cdot I + k_{r,a} \,, \tag{1a}$$

$$k_i(\lambda, I) = c_i(\lambda) \cdot I + k_{r,i} \,. \tag{1b}$$

The rates $k_{r,a}$ and $k_{r,i}$ describe thermal relaxation. However, the timescales used in the experiments are sufficiently small so that thermal relaxation can be neglected (**Smith et al., 2016**), i.e., we set $k_{r,a} = k_{r,i} = 0$. This implies that the conversion between the ON state and the OFF state of PhyB freezes when the system is switched to darkness.

The equilibrium constant of PhyB conversion, $K_L$, is given by the ratio of the conversion rates and depends only on the wavelength $\lambda$,

$$K_L(\lambda) = k_a(\lambda, I)/k_i(\lambda, I) = c_a(\lambda)/c_i(\lambda) \tag{2}$$

Thus, an altered light intensity affects the rates of PhyB conversion but leaves the equilibrium constant unchanged. The conversion dynamics of PhyB at concentration $P_{\mathrm{tot}}$ is described by the law of mass action. The PhyB ON concentration (denoted by $P_{\mathrm{ON}}$) is given by the following ordinary differential equation (**Figure 7—figure supplement 2A**),

$$\mathrm{d}P_{ON}/\mathrm{d}t = k_a(I)\,(P_{tot} - P_{ON}) - k_i(I)\,P_{ON} \,. \tag{3}$$

To keep the notation simple, the dependence of the conversion rates on the wavelength $\lambda$ has been dropped. The steady-state solution of **Equation (3)** depends only on $\lambda$ and the total PhyB concentration,

$$P_{ON,eq}(\lambda) = \phi_{ON,eq}(\lambda)P_{tot}. \tag{4}$$

Here we introduced the equilibrium fraction of PhyB ON,

$$\phi_{ON,eq}(\lambda) = K_L(\lambda)/(1 + K_L(\lambda)). \tag{5}$$

Thus changing the intensity at fixed wavelength does not impact the concentration of PhyB in the ON conformation.

The table below shows the equilibrium properties of PhyB conversion for the wavelengths used in the experiments (**Mancinelli, 1994**; **Smith et al., 2016**).

| $\lambda$ | $\phi_{a,eq}$ | $K_L$ |
|---|---|---|
| 660nm | 0.8 | 4 |
| 740nm | 0.01 | 0.01 |

# Binding model for PhyB interacting with GFP-PIF$^S$-TCR

Monomeric PhyB binds the GFP-PIF$^S$-TCR only in its active conformation (PhyB ON), while the inactive conformation does not interact with the GFP-PIF$^S$-TCR (for simplicity the GFP-PIF$^S$-TCR is called PIF-TCR in the remainder of the description of the mathematical model). We assume the total number of PhyB molecules to be in excess over the total number of PIF-TCR molecules to assure that the concentration of free PhyB is not depleted even if saturation is reached. We first consider the system kept in darkness (i.e., no conformational conversion of PhyB takes place) and PhyB ON at a fixed concentration $P_{ON}$ (**Figure 7—figure supplement 2B**). The rate of change of the concentration of PhyB ON-PIF-TCR complexes, denoted by $y_a$, is described by mass action kinetics,

$$\mathrm{d}y_a/\mathrm{d}t = k_{on} \cdot P_{ON} \cdot (1 - y_a) - k_{off} \cdot y_a \,. \tag{6}$$

The total amount of PIF-TCR on the surface of the T cell has been set to 1, i.e., **Equation (6)** yields the amount of PhyB ON-PIF-TCR complexes as fraction of all PIF-TCRs. The steady-state solution $y_{a,eq}$ is determined by the product of the binding affinity $K = k_{on}/k_{off}$ and the concentration $P_{ON}$,

$$y_{a,eq}(P_{ON}) = K \cdot P_{ON}/(1 + K \cdot P_{ON}) \,. \tag{7}$$

We now consider the interaction of PhyB and PIF-TCR in the presence of light of wavelength $\lambda$ and intensity $I$. If PhyB is in excess over PIF-TCR, the conversion dynamics of unbound PhyB is given by **Equation (3)**. Next to the formation and dissociation of PhyB ON-PIF-TCR complexes, light absorption allows for the transition of PhyB ON to PhyB OFF when PhyB ON is in complex with PIF-TCR (**Figure 7—figure supplement 2C**, left). In this state the complex can either dissociate with off-rate $k'_{off}$ or PhyB converts to its ON conformation with rate $k_a$. Importantly, the conversion rates of PhyB do not depend on whether PhyB is bound to PIF or not (**Figure 7—figure supplement 3**). The complex of PIF-TCR and PhyB OFF is very short-lived such that we assume $k'_{off} \gg k_a$. In this limit the amount of PhyB OFF bound to PIF-TCR can be neglected (**Figure 7—figure supplement 2C**, right), and the dynamics of the formation of PhyB ON-PIF-TCR complexes is given by (as above, the total amount PIF-TCR on the T cell has been set to 1),

$$\mathrm{d}y_a/\mathrm{d}t = k_{on} \cdot P_{ON} \cdot (1 - y_a) - \left(k_{off} + k_i(I)\right) \cdot y_a \,. \tag{8}$$

This is the same dynamics as in darkness (**Equation 6**) but dissociation is determined by an intensity-dependent effective off-rate,

$$k^*_{off}(I) = k_{off} + k_i(I) \,. \tag{9}$$

Thus, the PhyB-PIF system allows for tuning the off-rate in a linear manner by changing the light intensities (**Figure 7B**). Consequently, the affinity becomes intensity-dependent as well (**Figure 7C**),

$$K^*(I) = k_{on}/k^*_{off}(I) \,. \tag{10}$$

The effective affinity equals the affinity in **Equation (7)** if $I = 0$. It is the product of the effective affinity and the concentration of PhyB ON that determines the steady-state solution of **Equation (8)**,

$$y_{a,eq}(P_{ON}, I) = K^*(I) \cdot P_{ON}/(1 + K^*(I) \cdot P_{ON}) \,. \tag{11}$$

This is an important finding. Rather than just changing the rate of dissociation with light intensity, the amount of bound PIF-TCRs declines as well, which we confirmed experimentally (**Figure 7—figure supplement 4**). It is therefore critical to take this effect readily into account.

## Binding of oligomeric PhyB to PIF-TCR

To account for the oligomeric structure of the PhyB ligands used in the experiments, the concepts outlined above are generalized to account for multivalent binding. The principal difference between mono- and multivalent binding is found in the corresponding dose-response binding curves. While monovalent binding is a strictly increasing function of ligand concentration eventually reaching saturation, multivalent binding is characterized by a bell-shaped dose-response curve (**Perelson, 1981**). Indeed, our data are in line with this as the calcium response decreases at high concentrations (**Figure 6D**).

For simplicity we model the PhyB ligands as dimers, and we denote the concentration of PhyB oligomers with $j$ subunits in the ON conformation by $P_j$ ($j$ = 0, 1 and 2). At total concentration $P_{tot}$ (= $P_0 + P_1 + P_2$), the dynamics of PhyB conversion is given by

$$dP_1/dt = 2\,k_a(I)(P_{tot} - P_1 - P_2) - k_i(I)\,P_1 - k_a(I)\,P_1 + 2\,k_i(I)\,P_2 ~, \tag{12a}$$

$$dP_2/dt = k_a(I)\,P_1 - 2\,k_i(I)\,P_2 ~, \tag{12b}$$

$$P_0 = P_{tot} - P_1 - P_2 ~. \tag{12c}$$

The amount of PhyB ON subunits is $P_{ON} = P_1 + 2P_2$. The solution for the initial condition $P_0(0)=P_{tot}$, $P_1(0) = P_2(0)=0$ is given by

$$P_{ON}(t) = 2 \left[ 1 - e^{-(1+K_L)\,k_i(I)\,t} \right] \frac{K_L}{1+K_L}\,P_{tot} ~, \tag{13a}$$

$$P_2(t) = \left[ 1 - 2\,e^{-(1+K_L)\,k_i(I)\,t} + e^{-2(1+K_L)\,k_i(I)\,t} \right] \left( \frac{K_L}{1+K_L} \right)^2 P_{tot} ~, \tag{13b}$$

$$P_1(t) = P_{ON}(t) - 2\,P_2(t) ~, \tag{13c}$$

$$P_0(t) = P_{tot} - P_{ON}(t) + P_2(t) ~. \tag{13d}$$

The steady-state solution is

$$P_{ON,eq} = 2\,\frac{K_L}{1+K_L}\,P_{tot} ~, \tag{14a}$$

$$P_{1,eq} = \frac{2\,K_L}{(1+K_L)^2}\,P_{tot} ~, \tag{14b}$$

$$P_{2,eq} = \left( \frac{K_L}{1+K_L} \right)^2 P_{tot} ~, \tag{14c}$$

$$P_{0,eq} = \frac{1}{(1+K_L)^2}\,P_{tot} \tag{14d}.$$

This is accompanied by the binding dynamics. Next to the fraction of unbound PIF-TCR (= $T$), there are three possible binding states (**Figure 7—figure supplement 2D**): the PhyB ON subunit of $P_1$ being in complex with PIF-TCR (= $y_{1,1}$), one of the two PhyB ON subunits of $P_2$ being in complex with PIF-TCR (= $y_{2,1}$), and both PhyB ON subunits of $P_2$ each in complex with a PIF-TCR (= $y_{2,2}$). The corresponding rates of changes are described by mass action kinetics,

$$dy_{1,1}/dt = k_{on}\,P_1 \cdot T - \left( k_{off} + k_i(I) \right) y_{1,1} - k_a(I)\,y_{1,1} + k_i(I)\,y_{2,1} + 2\,k_i(I)\,y_{2,2} ~, \tag{15a}$$

$$dy_{2,1}/dt = 2\,k_{on}\,P_2 \cdot T - \left( k_{off} + k_i(I) \right) y_{2,1} + k_a(I)\,y_{1,1} - k_i(I)\,y_{2,1} - q_{on}\,y_{2,1} \cdot T + 2\,q_{off}\,y_{2,2} ~, \tag{15b}$$

$$dy_{2,2}/dt = q_{on}\,y_{2,1} \cdot T - 2\left( q_{off} + k_i(I) \right) y_{2,2} ~, \tag{15c}$$

$$1 = T + y_{1,1} + y_{2,1} + 2\,y_{2,2} ~. \tag{15d}$$

The last equation describes the conservation of total PIF-TCR, which, as above, we have set to 1. As a consequence, the binding rate of receptor cross-linking, i.e., PhyB tetramers binding bivalently to two TCRs, $q_{on}$ assumes the unit of an inverse time, making the cross-linking affinity $Q = q_{on}/q_{off}$ dimensionless.

Since there are two binding steps necessary to cross-link two PIF-TCRs, with each step potentially having its own off-rate, there are also two effective light-dependent off-rates,

$$k_{off}^*(I) = k_{off} + k_i(I), \tag{16a}$$

$$q_{off}^*(I) = q_{off} + k_i(I). \tag{16b}$$

Consequently, both affinities, the binding affinity from solution $K$ and the (dimensionless) cross-link affinity $Q$, are affected by the light intensity,

$$K^*(I) = k_{on}/k_{off}^*(I) , \tag{17a}$$

$$Q^*(I) = q_{on}/q_{off}^*(I) . \tag{17b}$$

To find the steady-state density of unbound PIF-TCRs, $T_{eq}(P_{ON}, I)$, we define $\xi(I) = k_i/k_{off}^*(I)$ and solve the following, effectively quadratic, equation for $T$,

$$1 = T + K^*(I) \cdot P_{ON} \cdot T + \frac{2 P_2 + K_L P_{ON} \xi(I)}{1 + [1 + K_L + Q^*(I) \cdot T] \xi(I)} Q^*(I) \cdot K^*(I) \cdot T^2 . \tag{18}$$

In the later section on model fitting, we need the steady-state amounts of cross-linked PIF-TCRs, $y_{X\text{-Link}} = 2 y_{2,2,eq}$, and bound PhyB ligands, $y_{bound} = y_{1,1,eq} + y_{2,1,eq} + y_{2,2,eq}$, which are given by,

$$y_{X-Link}(P_{ON}, P_2, I) = \frac{[2 P_2 + K_L P_{ON} \xi(I)] Q^*(I) \cdot K^*(I) \cdot T_{eq}(P_{ON}, I)^2}{1 + [1 + K_L + Q^*(I) \cdot T_{eq}(P_{ON}, I)] \xi(I)} , \tag{19a}$$

$$y_{bound}(P_{ON}, P_2, I) = 1 - T_{eq}(P_{ON}, I) - y_{X-Link}(P_{ON}, P_2, I) . \tag{19b}$$

## Kinetic proofreading of TCR-ligand binding by T cells

To carry out their immunological functions properly, T cells must reliably distinguish between foreign (i.e., stimulatory) and self (i.e., non-stimulatory) ligands, even if foreign ligands are buried in a large pool of self ligands. Several models exist in which the off-rate of the ligand-TCR interaction plays a key role in this discrimination (*Altan-Bonnet and Germain, 2005*; *Chakraborty and Weiss, 2014*; *Dushek et al., 2011*; *Lever et al., 2016*; *McKeithan, 1995*; *Rabinowitz et al., 1996*), and the underlying mechanisms are commonly referred to as kinetic proofreading (KPR). Although these models differ in the level of detail by which intracellular processes are accounted for, they share the property that downstream signaling is only initiated if the TCR can complete all processes the KPR mechanism comprises. For this to be accomplished, the ligand-TCR complex must sustain a TCR conformation/arrangement (*Gil et al., 2002*; *Martínez-Martín et al., 2009*; *Swamy et al., 2016*) that triggers the KPR processes sufficiently long; premature decay of this active TCR conformation leads to an instant reversion of all KPR modifications acquired so far. Thus, the KPR time $\tau_{KPR}$ sets a threshold for the half-life of the active TCR conformation $\tau_{active}$. It is this common aspect of the different KPR mechanisms that is the subject of interest in this work.

To quantify the effect of KPR on signal initiation, we take the probability of completing all KPR processes, $\alpha_{KPR}$, to be a strictly increasing function of $\tau_{active}$ that assumes its mid-value 0.5 if $\tau_{active} = \tau_{KPR}$. A convenient choice for such a function is given by

$$\alpha_{KPR}(\tau_{active}) = 2^{-\tau_{KPR}/\tau_{active}}, \tag{20}$$

for which $\alpha_{KPR} \to 1$ if $\tau_{active} >> \tau_{KPR}$, and $\alpha_{KPR} \to 0$ if $\tau_{active} << \tau_{KPR}$. Note that $\alpha_{KPR} = 1$ if $\tau_{KPR} = 0$, which reflects the absence of a KPR mechanism, because in this case the active TCR conformation initiates downstream signaling instantly.

The relation between the half-life of the active TCR conformation and the off-rate of the ligand-TCR interaction, $k_{off}$, and its half-life $\tau_D = \ln(2)/k_{off}$, depends on the way the TCR is stimulated. In the simplest case, ligands bind TCRs monovalently with this complex being already the active conformation (this might be the case with membrane-bound ligands). Then, $\tau_{active} = \tau_D$. However, the experimental setup used in this work requires cross-linking of TCRs, i.e., bivalent binding of the soluble ligand (*Figure 7—figure supplement 2D*), for inducing TCR signaling and thus establishing the active TCR conformation. Because the active TCR conformation of two cross-linked TCRs decays if one of the two subunits dissociates, see *Equation (15c)*, the half-life of the active TCR conformation is given by

$$\tau_{active}(I) = \frac{\ln(2)}{2\, q_{off}^*(I)} , \tag{21}$$

where *Equation (16b)* has been used. Note that *Equation (21)* is specific for soluble ligands and might assume a different form if ligands are membrane- or surface-bound.

## Example: Kinetic proofreading as proposed by McKeithan

To illustrate how an expression of the form *Equation (20)* arises in a KPR mechanism, we use the model proposed by McKeithan as an example (*McKeithan, 1995*). In this version receptors assuming an active TCR conformation are subject to sequential modifications each taking place with identical rates $k_p$ (*Figure 7—figure supplement 1A*). Downstream signaling is initiated only if $n$ modifications are completed. However, if the active TCR conformation decays at any stage with rate $k_{dec}$, all modifications acquired so far are lost rapidly.

The half-life of the active TCR conformation is given by $\tau_{active} = \ln(2)/k_{dec}$, and the time to acquire a single modification by $\tau_p = 1/k_p$. Since $n$ modifications are required for signaling, the average time from forming the active TCR conformation to establishing a signal is $\tau_{KPR} = n\,\tau_p$ (*Figure 7—figure supplement 1A*). The steady-state fraction of receptors in the active TCR conformation exhibiting all $n$ modifications is then given by

$$\alpha_n(\tau_{active}) = \left(\frac{\tau_{active}}{\tau_{active} + \frac{\ln(2)}{n}\tau_{KPR}}\right)^n . \tag{22}$$

We consider the case $n >> 1$, since the number of KPR processes involved in signal initiation is typically considered to be rather large. This gives

$$\alpha_n(\tau_{active}) \approx \left(1 - \frac{\ln(2)\tau_{KPR}}{n\,\tau_{active}}\right)^n . \tag{23}$$

In the limit of infinitely many modification steps, the KPR term becomes

$$\alpha_\infty(\tau_{active}) = 2^{-\tau_{KPR}/\tau_{active}}, \tag{24}$$

in agreement with *Equation (20)*. The KPR terms of McKeithan's version for different values of $n$ are shown in *Figure 7—figure supplement 1B*. The similarities between the curves shows that *Equation (24)* provides a good description independent of $n$.

## Determining the PhyB conversion dynamics

Prior to the exposure of the PhyB ligands to the T cells, nearly all PhyB subunits were initially in the OFF state, and the PhyB ligands were illuminated with a certain intensity of light of 660 nm wavelength to convert PhyB subunits to the ON state. This process is described by *Equations (13)*. For the binding and calcium measurements (*Figure 7—figure supplements 4* and *Figure 6D*), the duration of pre-illumination was chosen sufficiently long to reach equilibrium as given by *Equations (14)*. From the data at $I = 0$ of these calcium measurements, we can infer the relation between the calcium signal and the corresponding concentrations of the different PhyB forms in darkness, i.e., we consider *Equations (18)* and *Equations (19a)* for $I = 0$. Stopping the pre-illumination with intensity $I$ at some time point $T_{stop}$ before the equilibrium of PhyB conversion was reached trapped the concentrations of the different PhyB forms in a state that is described by *Equations (13)* evaluated at $T_{stop}$ and intensity $I$. The corresponding calcium response was again described by *Equations (18)* and *Equations (19a)* for $I = 0$ (because we switched to darkness after pre-illumination), but this time the PhyB concentrations did not assume their equilibrium values, *Equations (13)*. We can thus deduce the relation between $T_{stop}$ and the concentrations of the different PhyB forms for intensity $I$. Using different time points revealed the kinetics of PhyB conversion at the given intensity allowing for inferring the corresponding value of the conversion rate $k_i$. Repeating the protocol for a different intensity gives the same result but with changed rate $k_i$ according to *Equation (1b)*. Importantly, these results do not depend on the total PhyB concentration. This was indeed found experimentally (*Figure 8B*) and thus used to determine the conversion rates.

## T cell activation and model fitting

The central question of this work is whether T cells exploit a KPR mechanism. As shown above, the principal parameter of the KPR mechanism is its duration $\tau_{KPR}$, with $\tau_{KPR} = 0$ corresponding to no KPR mechanism being present. The active TCR conformation, which triggers a potential KPR mechanism and ultimately initiates downstream signaling, is given by TCRs bivalently bound by PhyB tetramers, i.e., cross-linked TCRs. The calcium response is thus taken to be proportional to the number of cross-linked TCRs,

$$Ca^{2+}(P_{ON}, P_2, I) \sim \alpha_{KPR}(\tau_{active}(I)) \cdot y_{X-Link}(P_{ON}, P_2, I) . \tag{25}$$

The KPR term is given by *Equation (20)*. The half-life of the active TCR conformation appearing in the KPR term of relation *Equation (20)* is given by *Equation (21)*. The cross-link term is determined by *Equations (18)* and *Equations (19a)*, and binding is described by *Equations (18)* and *Equations (19b)*. The total amount of PhyB in the ON state, $P_{ON}$, and the concentration of PhyB with both PhyB molecules in the ON state, $P_2$, are either determined by *Equations (13)* or by *Equations (14)*, depending on whether PhyB had been pre-illuminated with short pulses of 660 nm light to reveal the PhyB conversion kinetics (*Figure 8A and B*) or with an extended pulse to establish the PhyB equilibrium partition.

For the choice of the independent model parameters, it is convenient to re-define the conversion rate $k_i$, *Equation (1b)*, by

$$k_i(I) = k_{off} \cdot I/I_0 . \tag{26}$$

The parameter $I_0$ is the intensity at which the conversion rate $k_i$ equals $k_{off}$. We further define $\tau_D = \ln(2)/k_{off}$, the half-life of the PhyB ON-PIF-TCR complex in darkness, and $\gamma_{off} = q_{off}/k_{off}$ to quantify how distinct the off-rates, given by *Equation (16)*, are. With these definitions, the set of independent model parameters used for fitting is given in the following table:

| Parameter | Unit | Description |
| --- | --- | --- |

*continued on next page*

*continued*

| Parameter | Unit | Description |
|---|---|---|
| $K_D = k_{off}/k_{on}$ | nM | Dissociation constant of PhyB ON-PIF-TCR complex (darkness) |
| $Q = q_{on}/q_{off}$ | – | Association constant of multivalent binding step (darkness) |
| $\tau_{D,PhyB} = \ln(2)/k_{off}$ | s | Half-life of PhyB ON-PIF-TCR complex (darkness) |
| $\gamma_{off} = q_{off}/k_{off}$ | – | Distinction of off-rates of mono- and multivalent dissociation |
| $l_0$ | – | Intensity at which $k_i = k_{off}$ |
| $\tau_{KPR}$ | s | Duration of KPR mechanism (=threshold half-life of the ligand-TCR interaction) |

We fitted the models to the data using the maximum likelihood principle. For the binding data we used the estimated standard errors as obtained from the data. For the calcium signal we assumed an error model with a common coefficient of variation for all calcium data. All data (binding + all calcium measurements) were fitted simultaneously, providing a sufficiently large sample size (in total 130 data points).

To answer the question whether T cells exploit a KPR mechanism, we tested the hypothesis $\tau_{KPR} = 0$ against $\tau_{KPR} > 0$ by fitting the corresponding model to the data. Both fits were performed assuming $\gamma_{off} = 1$, i.e., we assumed that both off-rates were identical. A likelihood ration test revealed strong support for the alternative hypothesis $\tau_{KPR} > 0$ ($p<10^{-6}$, *Figure 7D*), showing the necessity to include a KPR mechanism in order to explain the data (*Figures 7E,F* and *8B*).

To this end, we assumed $\gamma_{off} = 1$. Because distinct off-rates imply a distinct dependence of the affinities *Equation (17)* on the intensity $I$, demanding $\gamma_{off} = 1$ if no KPR mechanism is considered might be too restrictive for explaining the data (if $\gamma_{off} \neq 1$ were actually the case), and a KPR mechanism could falsely be inferred. To avoid such a false-positive, we also considered the case $\gamma_{off} \neq 1$ without a KPR mechanism. This additional degree of freedom, however, lead to no improvement of the fit (*Figure 7—figure supplement 5*), which we take as further support for the existence of a KPR mechanism.

The parameter uncertainties of the model ($\tau_{KPR} > 0$ and $\gamma_{off} = 1$), expressed as 95% confidence intervals (95% CIs), were computed using the profile likelihood method (*Murphy and Van Der Vaart, 2000*; *Venzon and Moolgavkar, 1988*). All parameters were identifiable, i.e., have an upper and lower bound (*Figure 8—figure supplement 1*). This shows that the data were highly informative on the parameter values. The best-fit values and the lower and upper bounds of the 95% CIs are given in the table below:

| Parameter [unit] | Best-fit value | Lower bound | Upper bound |
|---|---|---|---|
| $K_D$ [nM] | 22 | 10 | 46 |
| $Q$ | 11 | 6 | 23 |
| $\tau_{D,PhyB}$ [s] | 40 | 15 | 150 |
| $l_0$ [% of max] | 2 | 1 | 3.5 |
| $\tau_{KPR}$ [s] | 8 | 3 | 19 |

A central assumption made in this study is that the modifications of the KPR mechanism are quickly removed upon dissociation. However, fast rebinding of the just dissociated ligand effectively increases the half-life sensed by a KPR mechanism (*Aleksic et al., 2010*; *Govern et al., 2010*). Following the arguments in *Govern et al. (2010)*, we calculate the expected number of rebindings in our system. The binding state triggering KPR in our model is two TCRs cross-linked by PhyB tetramers. Thus, if one TCR dissociates from PhyB, the other TCR remains bound and both TCRs become subject to complete modification removal unless quick rebinding occurs. Assuming the diffusivities of TCR and PhyB-bound TCR to be the same, the average number of rebindings, $N^*$, is given by (see *Equation (3)* in *Govern et al., 2010*),

$$N^* = \frac{q_{on}}{4\pi\, \mathrm{D}_{TCR}} \;, \tag{27}$$

with $D_{TCR}$ to diffusivity of TCRs. Using the best-fit value $q_{on}$ = 0.19 s$^{-1}$ and the values given in *Govern et al. (2010)* for the TCR diffusivity and TCR density, we find $N^*$=0.02. This shows that rebinding can be neglected in our system.

