## [Decision Letter]

Thank you for submitting your article "Optogenetic control shows that kinetic proofreading regulates the activity of the T cell receptor" for consideration by *eLife*. Your article has been reviewed by two peer reviewers, and the evaluation has been overseen by Arup Chakraborty as the Senior Editor and Reviewing Editor. The reviewers have opted to remain anonymous.

The reviewers have discussed the reviews with one another and the Reviewing Editor has drafted this decision to help you prepare a revised submission. Please aim to submit the revised version as a Tools and Resources paper (see comment below).

Summary:

The authors construct a light operated TCR using PhyB tetramers and a TCR modified with the PIF ligand and GFP. PhyB interaction with PIF is 100% turned off by far red light and 80% activated by red light, with the 20% off state in red light due to a probability of red light also turning the PhyB off. In the dark, the molecules maintain their state with a slow dark reversion of the off state. They trigger Jurkat cells expressing the PIF-TCR with the PhyB tetramers in on state and observe Calcium flux. They then note that 660 nm excitation drives an accelerated dissociation in proportion to intensity without changing the on-rate. They accelerate the dissociation with brighter red light to generate a dataset to test the kinetic proofreading model. The data and model suggest that in the conditions tested, the relevant time frame for interaction with the receptor that enables a calcium flux is 16 s.

This system has excellent properties for controlling kinetics using light. One might expect light to be used for spatial control, so its novel to apply this effectively to a problem in kinetic discrimination. Additional work would be needed to understand how this relates to selection thresholds, but together with the paper from the Weiner lab, this paper makes a strong case that optical control will be a powerful tool for understanding the role of kinetics in immunoreceptor signaling. As a technical paper that develops a tool, this is an interesting paper and could be published in the Tools and Resources category.

We do not think that the paper resolves the question of how kinetic proofreading occurs in any comprehensive way, which is what the authors set out to address. A number of technical points about the method need to be addressed. Furthermore, somewhat disturbingly, the vast literature in the field is cited in a selective manner. Below, we provide guidance on both these important issues.

Essential revisions:

1) The statements about affinity are too simplistic in the current understanding of the TCR-pMHC interactions. We suggest they replace affinity with potency (meaning functional potency).

2) When you introduce PhyB-PIF interaction rules you should state the affinity of the 100% on state. From PMID: 19749742 seems to be 20-100 nM in vivo. Would this allow them to load a cell with soluble PhyB and study the rate of dissociation, perhaps in a flow cell by microscopy? It would be fantastic to have a monomeric dissociation rate under different illumination conditions for the modeling.

3) Would it be possible to pre-load the tetramer to the cells to initiate a signaling condition and then illuminate to different intensities of 660 light to change the signaling state? It’s possible that already bound tetramers would remain bound even as the PhyB starts to cycle faster and this may result in a reset of signaling- loss with faster cycling, for example. Alternatively, could the authors link the PhyB to increase the valency such that the change in off-rate would not impact binding (due to greater rebinding). Perhaps a quantum dot or liposome decorated with multiple PhyB could be use employed. See: PMID: 17077145 or PMID: 9300688 for examples. These points need to be discussed.

4) The claim that you only affect half-life and no other parameter with the optogenetic method you employ is not supported by the data. The LOV2 system requires a conformational change to get the dissociation you need to end engagement (this is clearly stated in the text) and hence, you cannot know that the force of CAR interaction with the ligand is not changed in blue light when this conformational change takes place. Establishing this claim for no force change requires a direct test a la Chen Zhu's method to determine if this fundamental postulate of the method is true or not. This point needs to be discussed.

5) A number of relevant papers need to be cited that are pertinent, and the present results related to these studies. A small sampling of the missing literature are:a) The work from the Davis lab from several years ago on using caged pMHCs that were sensitive to light, which allowed measurements that described the speed at which signals propagate downstream.b) The role of rebinding that allows ligands with short half-lives to signal, described by work from eight years ago by Van der Merwe and co-workers (Immunity) and Huseby and co-workers (PNAS).c) The role of co-receptors, amount of active Lck and diffusion in kinetic proof reading described by Stepanek et al., 2014.d) You also ignore the data the Germain lab, and others, have presented showing that the same ab TCR signals differently in DP thymocytes vs. mature cells [in particular, allows meaningful propagation of signals for gene expression in DP with pMHC ligands incapable of electing the same response or biochemical changes in a mature T cell], meaning that intracellular events can control signal propagation separately from occupancy, half-life or mechanosensing.

6) The model for proofreading is a bit strange. Many people have made models that are far more realistic and incorporate feedback, including several papers from the Germain lab (Madrenas et al., 1995; Madrenas et al., 1997; Altman and Germain, 2005 and Nature. 2002 Nov 28;420(6914):429-34), Chakraborty and Palmer, Chakraborty and Weiss, and the more theoretical efforts by Murugan, Huse, and Leibler. These papers should at least be cited and briefly discussed.

7) The estimate of 3 proofreading steps is undoubtedly wrong. The various steps do not have the same time scales and there are feedback loops. These effects make the model's fit to obtain n very difficult to interpret. This part needs to be deleted.

---

## [Author Response]

Summary:

*The authors construct a light operated TCR using PhyB tetramers and a TCR modified with the PIF ligand and GFP. PhyB interaction with PIF is 100% turned off by far red light and 80% activated by red light, with the 20% off state in red light due to a probability of red light also turning the PhyB off. In the dark, the molecules maintain their state with a slow dark reversion of the off state. They trigger Jurkat cells expressing the PIF-TCR with the PhyB tetramers in on state and observe Calcium flux. They then note that 660 nm excitation drives an accelerated dissociation in proportion to intensity without changing the on-rate. They accelerate the dissociation with brighter red light to generate a dataset to test the kinetic proofreading model. The data and model suggest that in the conditions tested, the relevant time frame for interaction with the receptor that enables a calcium flux is 16 s.*

We thank you for this nice summary. Indeed, we found that in our system one individual ligand-TCR interaction has to have a half-life of 16 s for a half maximal response. However, we use a soluble ligand tetramer for which it has been shown that they need to bind at least bivalently to trigger the TCR (Boniface et al., 1998; Cochran et al., 2000; Minguet et al., 2007). In agreement with this, we showed that monomeric PhyB was not able to stimulate the GFP-PIF^S^-TCR (new Figure 5—figure supplement 1C). Thus, the relevant kinetic proofreading half-life is the one of bivalent binding and not of monovalent binding. For bivalent binding the relevant time frame is 8 s (modified Figures 8D and 9). We have elaborated on this in the new version and have changed the KPR time from 16 s to 8 s to avoid misunderstandings.

This system has excellent properties for controlling kinetics using light. One might expect light to be used for spatial control, so its novel to apply this effectively to a problem in kinetic discrimination. Additional work would be needed to understand how this relates to selection thresholds, but together with the paper from the Weiner lab, this paper makes a strong case that optical control will be a powerful tool for understanding the role of kinetics in immunoreceptor signaling. As a technical paper that develops a tool, this is an interesting paper and could be published in the Tools and Resources category.We do not think that the paper resolves the question of how kinetic proofreading occurs in any comprehensive way, which is what the authors set out to address. A number of technical points about the method need to be addressed. Furthermore, somewhat disturbingly, the vast literature in the field is cited in a selective manner. Below, we provide guidance on both these important issues.

We thank the reviewers for acknowledging the novelty of our work. As suggested we submit now as a Tools and Resources paper. We have extensively included discussion of the literature (see below).

Essential revisions:1) The statements about affinity are too simplistic in the current understanding of the TCR-pMHC interactions. We suggest they replace affinity with potency (meaning functional potency).

We agree with the reviewer that the term “affinity” is too simplistic, if it is the only one used to describe TCR-pMHC interactions. In particular, if more complex interactions – such as participation of co-receptors or multivalent binding – are to be summarized by a single parameter, terms like “potency” or “apparent affinity” are more appropriate. However, please note that in our case co-receptors are not present. Further, in our mathematical model, which is based on mass action kinetics and includes multivalent binding, we use the term “affinity” (denoted by K in the manuscript) exclusively to describe binding of PhyB to PIF-TCR from solution. (This is in line with the common usage of the term “affinity”, which refers to (the inverse of) the dissociation constant K_D_ of a binding reaction A+B ⇔ AB.) The formation of a bivalent binding state (i.e., two PIF-TCRs being simultaneously bound to the same PhyBt) is described by a “cross-link affinity”, denoted by Q in the manuscript. Thus, we describe equilibrium binding by two parameters, with the parameter K being in line with the common usage of the term “affinity”. Hence, we opted to keep our nomenclature of “affinity” and “cross-link affinity”. However, we included the following sentence when talking about pMHC-TCR and not PhyB-TCR interactions:

“Note that in case of pMHC binding to T cells other processes than the pure pMHC-TCR interaction are involved, such as interactions with the co-receptors CD8 or CD4; thus, the terms ‘apparent affinity’ or ‘potency’ might be more suitable when describing these complex binding events.”

2) When you introduce PhyB-PIF interaction rules you should state the affinity of the 100% on state. From PMID: 19749742 seems to be 20-100 nM in vivo. Would this allow them to load a cell with soluble PhyB and study the rate of dissociation, perhaps in a flow cell by microscopy? It would be fantastic to have a monomeric dissociation rate under different illumination conditions for the modeling.

The suggestion to measure the PhyB dissociation from our GFP-PIF^S^-TCR cells is well taken. Unfortunately, it was technically challenging for us to determine this dissociation rate experimentally.

Firstly, in order to detect monomeric PhyB, we cloned a chimeric protein comprising PhyB fused to the fluorescent protein mCherry. However, the recombinant expression of this chimeric protein in *E. coli* was very inefficient and yielded a variety of truncated proteins. Thus, we could not use this approach.

Secondly, in our lab we have not established a microscopic flow cell assay and therefore attempted to measure PhyB dissociation from the cells by flow cytometry by using the biotinylated, His_6_-tagged PhyB monomers. After fixation, we aimed to detect PhyB with fluorophore-coupled Streptavidin or with an anti-His_6_ antibody. However, in both cases the ratio of PhyB signal to background fluorescence was too low to reliably quantify the amount of bound PhyB. Note, that in the manuscript we have used PhyB tetramers for the flow cytometry stainings.

Thirdly, we quantified accumulation of dissociated PhyB in the cell supernatant by ELISA. These experiments were of low temporal resolution and did not provide an adequate dynamic range between the two extremes of maximally bound and maximally dissociated PhyB. Hence, we are not convinced that these results are of sufficient quality to include them in the manuscript. We think that the dissociation rate derived from the mathematical modelling of our datasets is a better estimation of the actual value than the aforementioned approaches. This is also reflected in the high similarity of our estimated value for the PhyB to PIF affinity (22 nM, 95% CI 10-46 nM) compared to published estimates (20-100 nM).

3) Would it be possible to pre-load the tetramer to the cells to initiate a signaling condition and then illuminate to different intensities of 660 light to change the signaling state? It’s possible that already bound tetramers would remain bound even as the PhyB starts to cycle faster and this may result in a reset of signaling- loss with faster cycling, for example. Alternatively, could the authors link the PhyB to increase the valency such that the change in off-rate would not impact binding (due to greater rebinding). Perhaps a quantum dot or liposome decorated with multiple PhyB could be use employed. See: PMID: 17077145 or PMID: 9300688 for examples. These points need to be discussed.

As suggested, we have pre-loaded the tetramer to the cells to initiate Ca^2+^ signaling and then illuminated with different intensities of 660 light to change the signaling state. We find that a faster cycling between the PhyB ON and OFF states (shorter half-life of binding) was able to stop a signal, similar as when we removed the ligand tetramer with 740 light. Interestingly, the threshold intensity of 660 light was the same as for induction of signaling, namely 3% that corresponds to a half-life of 8 s. This experiment is now included in new Figure 6—figure supplement 1B and C and described and discussed in the fourth paragraph of the subsection “The intensity of continuous 660 nm light determines GFP-PIF^S^-TCR activation”.

Since this experiment resulted in a clear cut result, we did not do experiments to change the valency of PhyB; except that we included a new experiment with PhyB monomers (new Figure 5—figure supplement 1C).

4) A number of relevant papers need to be cited that are pertinent, and the present results related to these studies. A small sampling of the missing literature are:a) The work from the Davis lab from several years ago on using caged pMHCs that were sensitive to light, which allowed measurements that described the speed at which signals propagate downstream.b) The role of rebinding that allows ligands with short half-lives to signal, described by work from eight years ago by Van der Merwe and co-workers (Immunity) and Huseby and co-workers (PNAS).c) The role of co-receptors, amount of active Lck and diffusion in kinetic proof reading described by Stepanek et al., 2014.d) You also ignore the data the Germain lab, and others, have presented showing that the same ab TCR signals differently in DP thymocytes vs. mature cells [in particular, allows meaningful propagation of signals for gene expression in DP with pMHC ligands incapable of electing the same response or biochemical changes in a mature T cell], meaning that intracellular events can control signal propagation separately from occupancy, half-life or mechanosensing.

We extended the Introduction and Discussion to discuss the literature more thoroughly as suggested (see also next point). We apologize for not having covered the literature well before. This was due to the extremely short article format we had chosen.

5) The model for proofreading is a bit strange. Many people have made models that are far more realistic and incorporate feedback, including several papers from the Germain lab (Madrenas et al., 1995; Madrenas et al., 1997; Altman and Germain, 2005 and Nature. 2002 Nov 28;420(6914):429-34), Chakraborty and Palmer, Chakraborty and Weiss, and the more theoretical efforts by Murugan, Huse, and Leibler. These papers should at least be cited and briefly discussed.

In the Introduction section we have now discussed the work where kinetic proofreading models included feedback and feed-forward loops:

“The KPR model has also been extended to include feedback and feed-forward loops in the signaling network below the TCR (Altan-Bonnet and Germain, 2005; Chakraborty and Weiss, 2014; Dushek et al., 2011; Lever et al., 2016; Rabinowitz et al., 1996). […] At the same time, the high sensitivity of the T cells towards low numbers of ligands (1-10 molecules) was retained (Irvine et al., 2002; Purbhoo et al., 2004).”

6) The estimate of 3 proofreading steps is undoubtedly wrong. The various steps do not have the same time scales and there are feedback loops. These effects make the model's fit to obtain n very difficult to interpret. This part needs to be deleted.

Our conclusion was that *at least* 3 proofreading steps have to occur and not that 3 steps occur. However, we agree with the reviewer that the KPR model underlying the estimate of proofreading steps (McKeithan, 1995) is too simplistic, and we have thus deleted our conclusion about the number of proofreading steps.